# Large scale automated phylogenomic analysis of bacterial isolates and the Evergreen Online platform

Judit Szarvas [1,2 ✉], Johanne Ahrenfeldt [1,2], Jose Luis Bellod Cisneros[1], Martin Christen Frølund Thomsen[1], Frank M. Aarestrup [1] & Ole Lund [1]

Public health authorities whole-genome sequence thousands of isolates each month for microbial diagnostics and surveillance of pathogenic bacteria. The computational methods have not kept up with the deluge of data and the need for real-time results. We have therefore created a bioinformatics pipeline for rapid subtyping and continuous phylogenomic analysis of bacterial samples, suited for large-scale surveillance. The data is divided into sets by mapping to reference genomes, then consensus sequences are generated. Nucleotide based genetic distance is calculated between the sequences in each set, and isolates are clustered together at 10 single-nucleotide polymorphisms. Phylogenetic trees are inferred from the non-redundant sequences and the clustered isolates are added back. The method is accurate at grouping outbreak strains together, while discriminating them from non-outbreak strains. The pipeline is applied in Evergreen Online, which processes publicly available sequencing data from foodborne bacterial pathogens on a daily basis, updating phylogenetic trees as needed.

---

[1] Research Group for Genomic Epidemiology, National Food Institute, Technical University of Denmark, Kongens Lyngby, Denmark. [2] These authors contributed equally: Judit Szarvas, Johanne Ahrenfeldt. ✉email: jusz@dtu.dk

Epidemiological typing of bacteria is used by hospitals and public health authorities, as well as animal health authorities, to detect outbreaks of infectious diseases and determine trends over time. Traditionally, this includes culturing and isolating the pathogen, followed by species identification and subtyping using various conventional microbiological and molecular methodologies.

For outbreak investigation, however, it is necessary to place the infectious agent into a more discriminatory category than species, to establish links between cases and sources. Multi-locus sequence typing (MLST) has been a frequently used molecular subtyping method, where sequence types are assigned to the isolates based on the combinations of alleles for 6–10 housekeeping genes[1].

Whole-genome sequencing (WGS) has opened a new chapter in microbial diagnostics and epidemiological typing. WGS data can be used to determine, amongst other characteristics, both MLST types and serotype of several bacterial species[2,3]. Furthermore, several studies for multiple bacterial species have shown the value of WGS for elucidating the bacterial evolution and phylogeny, and identifying outbreaks[4–6].

The use of WGS has enabled the unbiased comparison of samples processed in different laboratories, boosting surveillance and outbreak detection, but the methods for sharing and comparing a large number of samples have not been established yet[7,8]. Therefore, a number of national, regional, and international initiatives have been launched with the aim of facilitating the sharing, analyses and comparison of WGS data[9–11].

Since 2012, the US Food and Drug Administration (FDA) is leading a network of public health and university laboratories, called GenomeTrakr. These laboratories sequence bacterial isolates from food and environmental samples, and upload the data to the National Center for Biotechnology Information (NCBI). GenomeTrakr is restricted to foodborne pathogens and currently includes data from seven such bacterial species[12]. All raw WGS data are publicly shared through NCBI, facilitating the collaboration between laboratories. Furthermore, the raw data are picked up by the NCBI Pathogen Detection pipeline[13], that assembles the samples into draft genomes to predict the nearest neighbors and construct phylogenetic trees for each within-50-SNPs cluster using an exact maximum compatibility algorithm[14]. This approach requires access to all of the raw data or assembled genomes, and very extensive computational resources for larger databases, like *Salmonella enterica*. In addition, no sub-species taxonomical classification has so far been implemented in the pipeline.

Focusing on the same bacterial species as GenomeTrakr, PulseNet USA has also established procedures for use of WGS data for outbreak detection. In their vision, an extension of the highly successful MLST approach into a core-genome (cgMLST) or whole-genome (wgMLST) scheme, with in the order of a thousand genes, would allow for sharing information under a common nomenclature[11]. MLST schemes are offered from several databases[15–17], and a number of, at times conflicting, cg- and wgMLST schemes have recently been proposed for a limited number of bacterial species[16,18–24]. Moreover, few of the proposed schemes provide a definitive nomenclature of sequence types to go with the allele profiles. The existing schemes do not cover all of the potential allelic variation: a recent study showed, that for *Campylobacter jejuni*, that has maintained MLST schemes, only approximately 53% of the strains of animal origin could be assigned to an existing unique allelic profile[25]. Continuous curation of the hundreds of relevant bacterial species, that are known human, animal and plant pathogens, would require great effort. A centralized database for the distribution of the allele profiles and sequences would be also necessary. Furthermore, for comparable results, and surveillance, the same analysis

pipeline or software should be used for the prediction of the allelic profiles. For example, single-linkage cgMLST clusters can be generated of public and private uploaded data on EnteroBase[15], and up to 1000 sequences on Pathogenwatch[26], by manual selection of strains to be included in the analysis.

The results generated by gene based approaches often lack the necessary resolution, and in most cases, selected WGS data are further analyzed using single nucleotide profiling. Here, genomic variants (single nucleotide polymorphisms (SNPs), insertions and deletions) are derived by aligning WGS reads to a reference genome. For each bacterial species, custom single nucleotide profiling (SNP validation, cluster threshold determination, etc.) is necessary in order to achieve results that are biologically relevant and informative. Here, the choice of the reference genome is crucial to maximize the number of SNPs that could be detected, as SNPs outside the regions covered by the reference are overlooked. The analyzed samples and the reference genome are chosen on a case-by-case basis, usually based on the subtyping results. Various offline SNP analysis pipelines are used by laboratories and research groups for inferring phylogenetic trees for isolates of interest[27–32]. For example, Public Health England developed and uses SnapperDB for outbreak detection without initial cluster analysis by cg- or wgMLST. SnapperDB consists of tools to create a database of SNPs compared to a given reference sequence, and assign each isolate a SNP address based on single linkage clustering[33]. As for web based solutions, in addition to cgMLST clustering, EnteroBase also offers SNP analysis of user selected strains based on the predicted genotypes[15]. Real-time tracking of bacterial pathogens with nucleotide resolution is performed by NCBI Pathogen Detection platform, that processes WGS data from selected bioprojects[13]. For viral pathogens, real-time tracking and visualization of evolution has been realized for seasonal influenza with Nextflu[34], and generalized in Nextstrain[35].

We present here a whole-genome, SNP-based method for subtyping and preliminary phylogenomic analysis of bacterial isolates, that circumvent the known limitations of current gene- and SNP-based approaches. PAPABAC carries out rapid and automated subtyping of bacterial whole-genome sequenced isolates and generates continuously updated phylogenetic trees based on nucleotide differences. We demonstrate two applications, a standalone version for local monitoring of bacterial isolates, and Evergreen Online, for global surveillance of foodborne bacterial pathogens. We also suggest a stable naming scheme for each isolate, making the results from the pipeline easier to communicate to others. To the best of our knowledge, no such tool exists at the moment.

## Results

**Automated phylogenomic analysis of bacterial WGS data**. We developed PAPABAC (Fig. 1), a pipeline for automated phylogenomic analysis of bacterial isolates, that needs no additional input besides WGS data (FASTQ files) and generates clusters of closely related isolates. PAPABAC first matches the isolates to complete bacterial chromosomal genome reference sequences with greater than 99.0% sequence identity and a minimum average depth of 11. These reference sequences serve as templates for the alignment of the raw reads. The aligned bases at each position are statistically evaluated to determine the consensus sequence, as previously described for a nucleotide difference method[36]. Positions that do not fulfill the significance criteria remain ambiguous, get assigned "N", and are disregarded during the pairwise genetic distance calculation. These steps ensure that there is high confidence in the consensus sequence that is the basis of the genetic distance estimation.

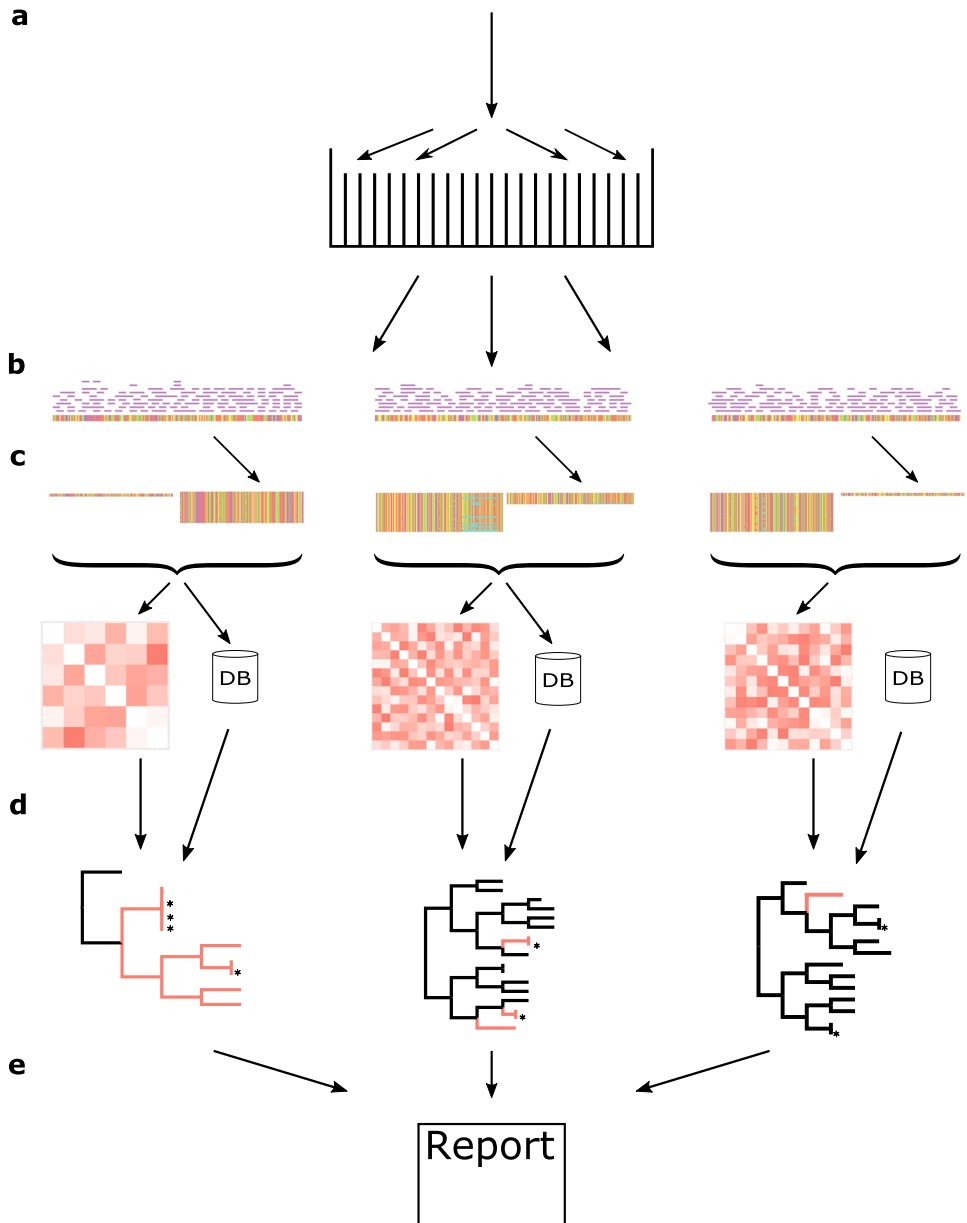

**Fig. 1 Overview of PAPABAC. a** The input raw read files are classified into sets based on k-mer similarity to NCBI RefSeq complete prokaryotic chromosomal genomes. **b** The raw reads are mapped to the reference genome and a consensus sequence is generated via strict statistical evaluation ($p < 0.05$) of the mapped bases in each position. **c** The resulting consensus sequences are of equal length in each template set. The new isolates in each set are clustered to the non-redundant isolates already in the set if the pairwise nucleotide difference based genetic distance is less than 10. The remaining new isolates undergo the same clustering process. **d** Pairwise genetic distance between all non-redundant isolate in the set is used as input for neighbor-joining algorithm. If there are less than 600 non-redundant isolates in a set, an approximately maximum likelihood phylogenetic tree is also inferred based on the consensus sequences (red: new isolates). The clustered isolates are placed back onto the trees with 0 distance to the cluster representative (marked with an asterisk). **e** The information about the acquired isolates, the sets, the clusters, and the phylogenetic trees is stored in SQLite databases, which are queried once all sets with new isolates are processed to output the results to the users.

The pipeline retains analysis results in such a manner that input is added to the previously processed data. The phylogenomic analysis is carried out on the current input and the previously found non-redundant isolates (singletons and cluster representatives). The genetic distance is estimated in a pairwise manner, comparing the given two sequences for all non-ambiguous positions, i.e., positions where none of the two sequences have an "N" assigned. The distances between the previously processed runs are stored on disk, saving computational time, and only the distances to the new isolates are computed in a given run.

A clustering step during the genetic distance calculation forms clusters of closely related isolates and reduces the number of similar sequences in each set, and thereby also reduces the computation time. After identifying a non-redundant isolate and a closely related isolate to it, the one previously deemed non-redundant will be the cluster representative and kept, while the clustered one is omitted from the subsequent runs of the pipeline. However, the information about the clustering is added to a database and the clustered isolate will be placed on the inferred phylogenetic tree. The cluster representatives remain constant through the subsequent runs of the pipeline, and the clusters only

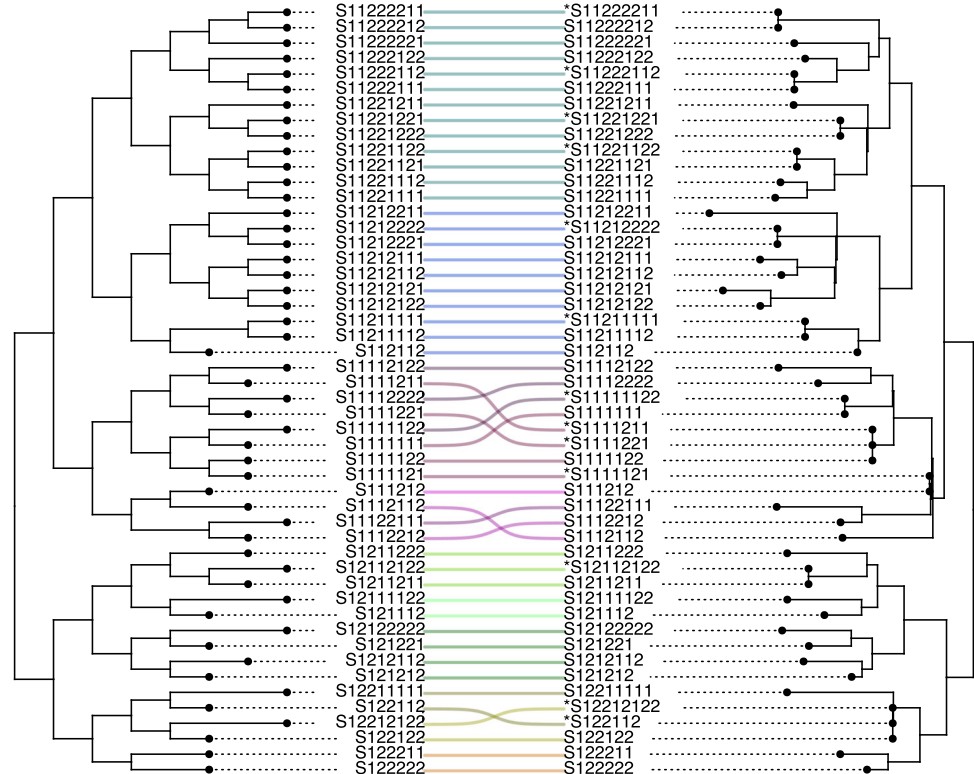

**Fig. 2 Benchmarking of PAPABAC.** Comparison of the ideal tree (left) to the PAPABAC maximum likelihood tree made of the in vitro experiment dataset[36]. Taxa with an asterisk were clustered together with the taxa in the same clade.

increase in size if new isolates are clustered with the representative. Therefore, each cluster is stable in the sense that an isolate will never change which representative it is associated with and each cluster can be reliably identified by the template name and the identifier of its cluster representative.

The pipeline can be run on a computer with 8 Gb RAM and Unix system. The computational time is reduced compared to re-running the whole analysis each time new samples are added, even without parallelization (Supplementary Fig. 1).

PAPABAC was benchmarked against three SNP pipeline benchmarking datasets. An *Escherichia coli* in vitro evolution experiment dataset[37] provided 50 closely related samples on a short temporal scale with less than 100 nucleotide differences across the dataset. The algorithm clustered together seven out of ten samples with the same ancestor that were taken on the same day and presumably had less than ten nucleotide differences between them. The PAPABAC maximum likelihood (Fig. 2) and neighbor-joining (Supplementary Fig. 2) trees with the clustered isolates pruned to resolve the polytomies were comparable to the ideal phylogeny of the in vitro experiment dataset: the normalized Robinson-Foulds distances were 0.18 and 0.12, respectfully. Benchmarking against the *Campylobacter jejuni* (Supplementary Fig. 3a, b) and the *Listeria monocytogenes* (Supplementary Fig. 3c, d) datasets from Timme et al.[38], PAPABAC correctly clustered the related outbreak strains (colored) and the outgroups, where the genetic distance was below the clustering threshold. The topologies of the maximum likelihood phylogenetic trees closely resembled the tree topologies given.

**Online surveillance of foodborne bacterial pathogens.** Evergreen Online was built on PAPABAC. Raw WGS data files of five major foodborne pathogens (*C. jejuni, E. coli, L. monocytogenes, Salmonella enterica*, and *Shigella* spp.) are downloaded daily from public repositories with the aim of global surveillance of potential outbreaks worldwide. The inferred phylogenetic trees and information about all of the isolates in the system are available and searchable on the website (http://cge.cbs.dtu.dk/services/Evergreen). The full phylogenetic trees can be viewed in the platform, or in Microreact[39], where the temporospatial information of the samples are also presented visually. Recognizing that phylogenies with more than a few hundred nodes are difficult to browse, the hyperlink of the cluster representative leads to subtrees, limiting the taxa shown to those surrounding the given cluster. Refined phylogenetic trees, based on SNP differences between all isolates around the cluster representative, are also published in Evergreen Online. The platform has been available since October 1st 2017, with logs reliably saved since October 28th 2017. The number of raw read files downloaded fluctuates with the work week of the public health laboratories. On busier days, more than 800 isolates are downloaded. The average number of isolates downloaded per day is 418. Downloading and mapping to the reference genomes take 130 min on average, with the majority of the time spent on downloading. Alignment of the raw reads and the generation of the consensus sequences takes on average 9 min per isolate. The computing time for the template sets is dependent on the number of non-redundant and new sequences in each set, but in most cases even the slowest is finalized within 5 h (Fig. 3 and Supplementary Data 1).

As of June 26th 2018, the pipeline downloaded 82,043 isolates. Out of these, 63,276 isolates have been mapped to references with at least 99.0% identity and average depth of 11 (Supplementary Fig. 4a). The majority of the isolates were typed as *Salmonella enterica* (59.1%), followed by *Escherichia coli* (19.4%) (Supplementary Fig. 4b). The two largest template sets are *S.* Dublin and *S.* Typhimurium serovars, with both close to 9500 isolates in total. After the homology reduction there were 3216 and 5093 non-redundant sequences in these sets, respectively. As of July 13th

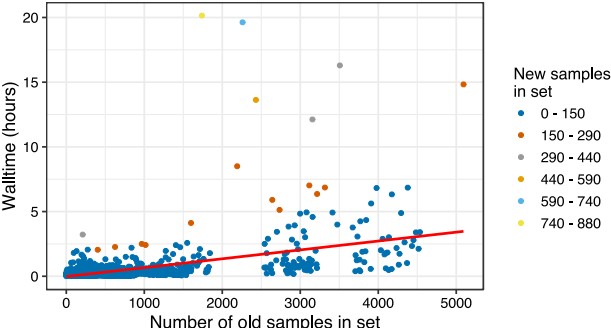

**Fig. 3 Walltime of the phylogenetic analysis.** Time requirement of the phylogenomic analysis for given number of non-redundant and new strains, on 20 CPUs.

2018, on average, 67% of the sequences are non-redundant in the template sets, while the E. coli template sets are the most diverse and the *Listeria monocytogenes* ones are the least diverse (Supplementary Fig. 4c). There were 122 isolates predicted to have a type not specified by the query (Supplementary Table 1). Of these, 14 isolates were mixed samples, composed of both the queried and the non-queried organisms.

The *L. monocytogenes* SNP pipeline benchmarking dataset[38] was added to the template set (Listeria_monocytogenes_07PF0776_NC_017728_1) of the corresponding reference genome in Evergreen Online, to test the sensitivity and accuracy of the clustering in large datasets. This template set at that moment contained more than 2400 isolates, of which 1398 were non-redundant. The isolates were placed onto a clade of a clonal lineage. The outbreak and outgroup isolates were separated in concordance with the ideal phylogeny (Fig. 4). The smaller clade of outbreak samples clustered to a sample (SRR538386) of an environmental swab in 2014, from California, USA.

Isolates that were presumed to be from an *E. coli* O157:H7 outbreak were selected for the comparison of Evergreen Online and the NCBI Pathogen Detection platform (NCBI-PD). They were located on the Escherichia_coli_O157_H7_str_Sakai_chromosome_NC_002695_1 neighbor-joining tree from Evergreen Online and the PDS000000952.271 SNP cluster tree from NCBI-PD. The labeled isolates appeared in three clusters on the neighbor-joining tree. There were 19.9 nucleotide differences between the yellow and the red cluster representatives and 12.6 nucleotide differences between the yellow and the blue cluster representative. On the NCBI-PD tree, the isolates marked with red circles were on the same clade, while the ones marked with blue and yellow were intermixing on clades that were, at most, 15 compatible characters apart (Fig. 5). On the refined subtree encompassing the labeled isolates, the yellow and blue labeled isolates intermix, similarly to the NCBI-PD tree (Supplementary Fig. 5).

## Discussion

Whole-genome sequencing, performed alongside the traditional methods in routine microbiology, yields hundreds to thousands of WGS isolates yearly in hospital, public health and food safety laboratories. This amount of data is overwhelming for many, and there is a lack of methods to generate a quick overview and help prioritize resources. The timely analysis of the sequencing data would allow the detection of more bacterial outbreaks and aid the prevention of further spread. However, lack of human and computational resources for this demanding task often hampers the prompt analysis of the data. Automating the initial subtyping phase would facilitate the start of an outbreak investigation. PAPABAC offers rapid subtyping for a wide range of prokaryotic

organisms: the supplied database covers all bacterial subtypes with complete genomes present in NCBI RefSeq. Further reference genomes could be added to increase the covered sequence space, but the active curation of the reference database is not required for routine use. The selection of the reference sequence for the phylogenomic analysis is fast and robust. It is independent of pre-assumptions about the isolates: misclassification during previous analysis does not introduce errors into the downstream analysis. Contamination from another species is discarded during the consensus sequence generation. The subtyping step via k-mer based mapping to a close reference also serves as a sequencing quality control measure, because low-quality sequencing runs will typically result in isolates with low identity to any reference and/ or low depth. These isolates do not progress further to the phylogenomic analysis, as they would not yield reliable results.

The phylogenomic analysis performed on the template sets has higher discriminatory power than cg- or wgMLST. The underlying nucleotide difference method was validated in five different studies[6,36,37,40,41]. By using all positions in the consensus sequences for estimating the genetic distance, instead of considering only selected loci, we ensure a high level of sensitivity, as we also include mutations that occur between genes.

The clustering step during the genetic distance calculation was introduced in order to reduce the homology in the template sets and thus reduce the computational burden as the template sets increase in size. However, the clustering threshold of 10 nucleotide differences also constructs informative clusters of highly similar isolates. Benchmarking with the *E. coli* in vitro evolution experiment dataset (Fig. 2) showed that the algorithm was capable of correctly clustering isolates that were derived from the same ancestor, while distinguishing them from other closely related strains. The same sensitivity was demonstrated on empirical outbreak datasets (Supplementary Fig. 3), where the pipeline clustered the outbreak-related strains and separated them from the outgroup strains. Both the maximum likelihood inferred and the neighbor-joining trees placed the outbreak strains correctly in the phylogeny. These results show, that PAPABAC provides quick and reliable information about the close relatives of an outbreak strain to provide candidates to perform a more thorough analysis on.

The design of PAPABAC means that once an isolate passed the homology reduction step, it will be present in the subsequent runs of the pipeline. When an incoming isolate is highly similar to a non-redundant one, the more recent will be the one that is clustered, added to the database and removed from further runs. Hence, the cluster representatives and clusters are robust to the addition of new data to the analysis. Therefore, PAPABAC yields a stable and communicable name for the clusters, comprised of the template name and the cluster representative. This is an advantage over cg- and wgMLST, where allelic profiles don't necessarily have communicable names, and the clusters could merge.

Evergreen Online has been steadily processing WGS data of foodborne bacterial pathogen isolates collected worldwide in real time (Supplementary Fig. 4a). It has been able to keep pace with the flow of the generated data that mainly came from public health and food safety laboratories. Excluding the download time and the optional maximum likelihood based phylogenetic inference, the whole analysis is done in less than a day, even for template sets with thousands of isolates (Fig. 3). This turnover time facilitates quick response in a potential outbreak scenario.

The isolates are not distributed equally across the templates in the system (Supplementary Fig. 4b). Out of the five queried species, *S. enterica* isolates are disproportionally represented. Sequences in the *S. Dublin* and the *S. Typhimurium LT2* template sets comprise in total approximately half of the *S. enterica*

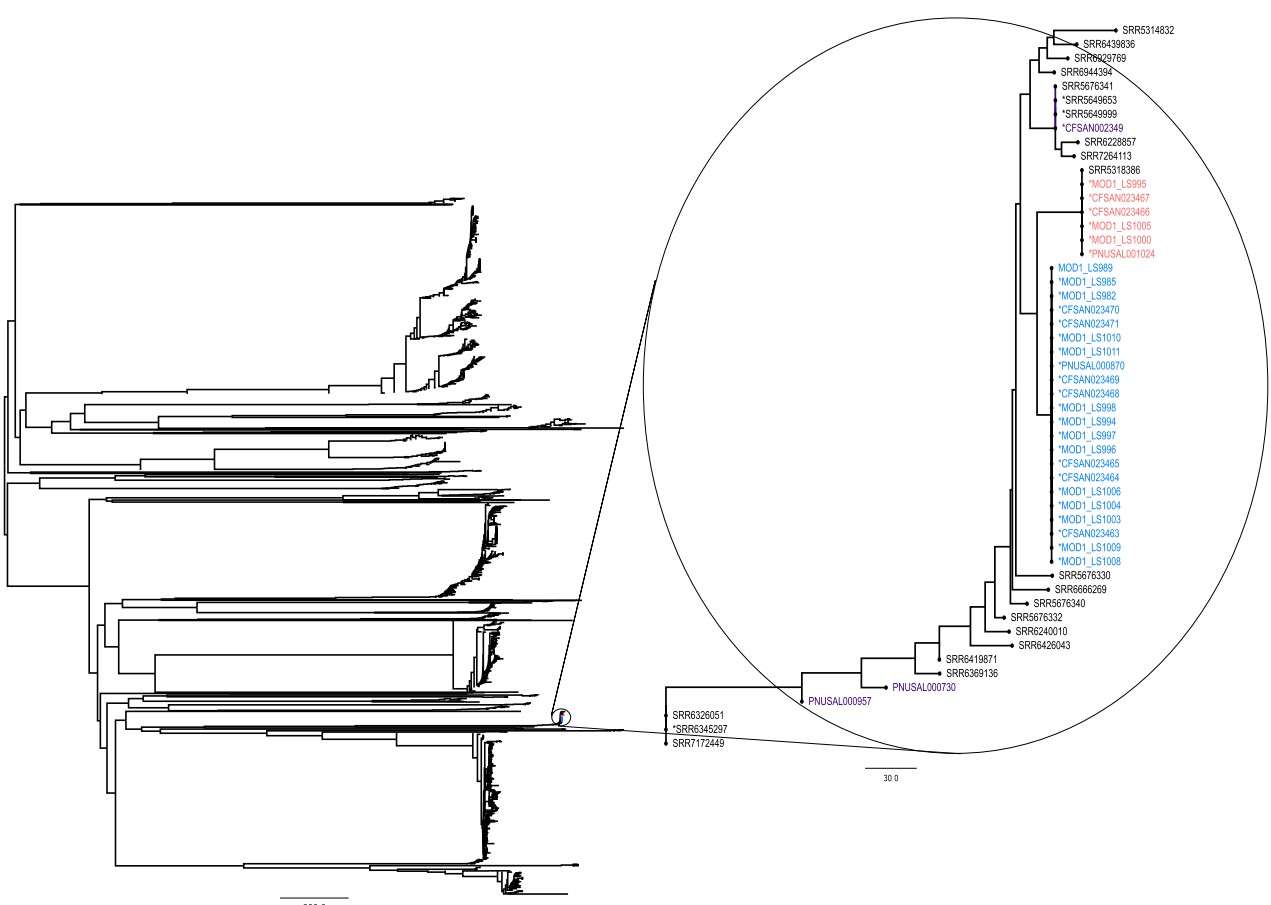

**Fig. 4 Benchmarking of Evergreen Online on benchmark dataset for phylogenomic pipelines.** Neighbor-joining tree for the Listeria_monocytogenes_ 07PF0776_NC_017728_1 set after the samples of the *L. monocytogenes* dataset were added. The red and blue isolates belong to the two original outbreak clades, and the purple isolates are outgroups in the benchmark dataset.

isolates. The sequence diversity in the template sets is varied, but the homology reduction on the template sets reduces the number of sequences approximately by a third, considerably decreasing the computational time. The *L. monocytogenes* template sets were the least diverse, which could be due to sampling bias: bacteria that are present in the environment are routinely sampled from food production sites multiple times, producing highly similar sequences, that are then removed from the ongoing analysis. We also tested how a large number of sequences already present in a template set would affect the ability of the pipeline to discriminate between samples (Fig. 4). The template set that corresponded to the stone fruit *L. monocytogenes* outbreak dataset reference had more than 1000 non-redundant isolates, which was ideal for the test analysis. The isolates that were part of the same outbreak clustered together and formed the two expected outbreak clusters, despite the confounding presence of the sequences already in the template set. The smaller clade, however, had a different cluster representative when using all data for the template set, compared with analysis of the outbreak data alone: an environmental sample, that could be related to the outbreak, as it was sampled from the same US state and year (California, 2014) as the samples in the outbreak dataset. These findings indicate that the pipeline is capable of identifying closely related samples, however it is necessary to conduct epidemiological analysis and apply other knowledge when interpreting the results.

Evergreen Online allows for automated selection of closely related isolates out of thousands, which is also the objective of NCBI-PD. *E. coli* isolates, situated on three clusters in Evergreen Online and supposedly from an outbreak, were located in NCBI-

PD and their placement in the SNP cluster tree was compared to the Evergreen Online tree (Fig. 5). One cluster (red) was in agreement between the two platforms, and samples from the other two (yellow and blue) clusters were intermixing on a clade on the NCBI-PD tree. The nucleotide difference counts between these samples are low and the differences between the phylogenomic methods could lead to differences in the finer details of the inferred phylogenies. On the refined subtree (Supplementary Fig. 5), the labeled samples form similar topology to the NCBI-PD tree. The homology reducing clustering in Evergreen Online means that any sample in the cluster is less than ten nucleotides differences from the cluster representative, however, the differences between the samples could amount to 18 nucleotides. The compatible character distances on the NCBI-PD tree between the mixed samples are less than that. Taking this into account, the observed distribution of the labeled samples in the two platforms are concordant.

In summary, we developed PAPABAC with the aim of rapid subtyping and continuous phylogenomic analysis on a growing number of bacterial samples. PAPABAC overcomes limitations of cg- and wgMLST approaches by tolerating genomic variation during subtyping, but providing greater sensitivity during the phylogenomic analysis. It was benchmarked on datasets created for testing SNP-based pipelines, and was proved to be accurate in discriminating between outbreak related and non-related samples. The software is open source and fulfills expectations put to WGS-based surveillance pipelines (Table 1). Evergreen Online, an application made for the global surveillance of foodborne bacterial pathogens, demonstrates the accuracy, speed, stability and

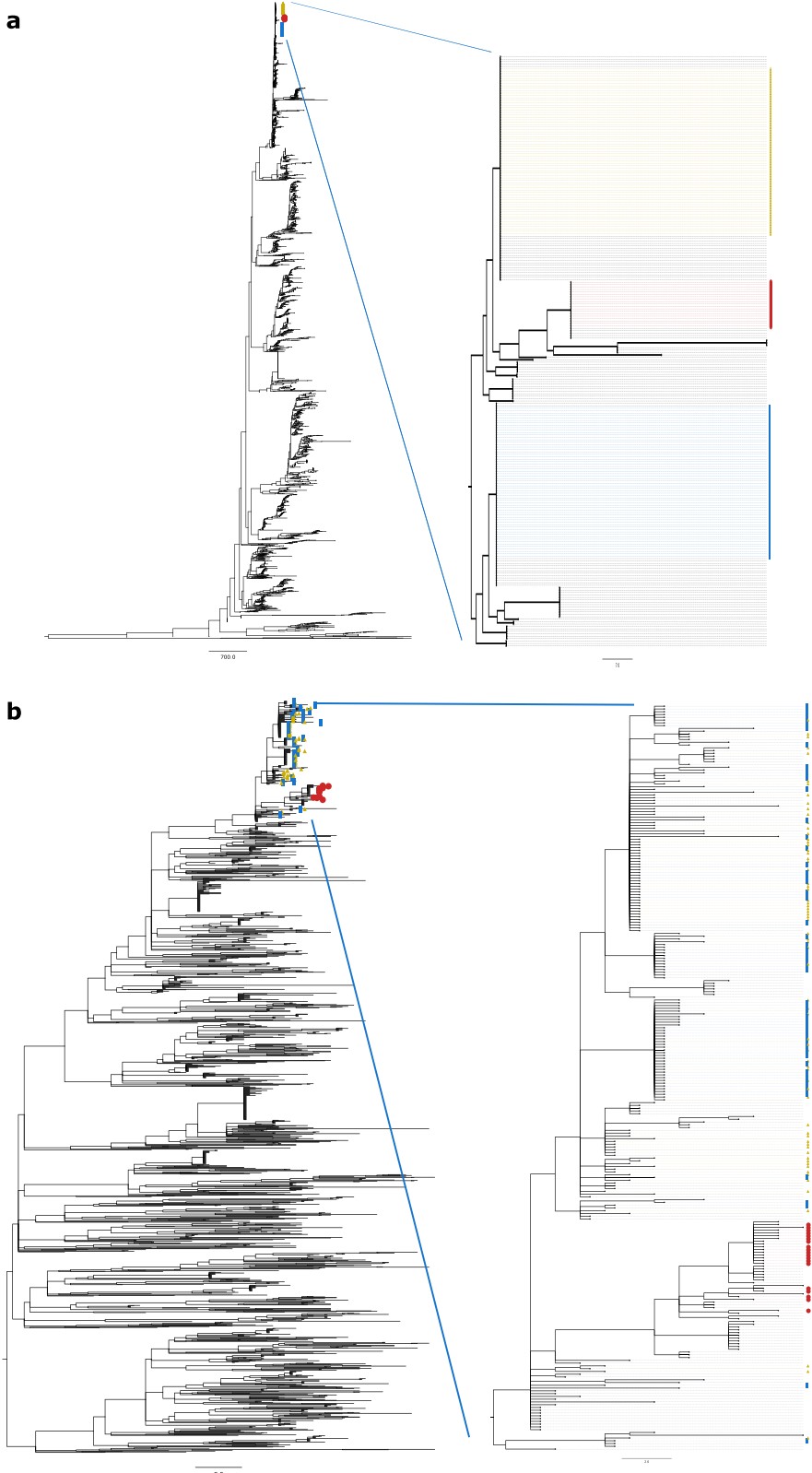

**Fig. 5 Comparison of Evergreen Online and the NCBI Pathogen Detection platform.** Selected isolates in the **a** Escherichia_coli_O157_H7_str_Sakai_ chromosome_NC_002695_1 NJ tree and on the **b** PDS000000952.271 SNP cluster maximum compatibility tree. The three largest clusters of the selected samples on the NJ tree are labeled with yellow, red, and blue dots. These isolates were marked with the same labels on the NCBI-PD tree. The red labeled ones are on a single clade on the PD tree, while the blue and yellow isolates are mixing on two other clades.

| Table 1 Comparison of pipelines for large-scale surveillance for pathogenic bacteria. | | | |
|---|---|---|---|
| | **SnapperDB** | **NCBI-PD** | **PAPABAC** |
| For a wide range of bacterial species | x | x | x |
| Only sequencing data is required input | — | x | x |
| Whole-genome based | x | x | x |
| Assembly-free | x | — | x |
| Quality control steps | x | x | x |
| Automated phylogenomic analysis | — | x | x |
| Stable clustering of samples across runs | — | — | x |
| Communicable nomenclature for subtype and cluster | x | — | x |
| Open source | x | — | x |

practicality of PAPABAC on thousands of samples via an ongoing analysis, where the results are published online.

## Methods

**PAPABAC pipeline**. The pipeline takes raw whole-sequencing reads (FASTQ files) as input. Matching reference sequences (templates) in our reference database, that have greater than 99.0% identity and a minimum average depth of 11, are identified for the isolates using 16-mers via KMA[42] in sparse mode. Multiple templates are accepted, if they meet the criteria, allowing for the procession of mixed samples. Information about the runs and their templates are inserted into the main SQLite database. The isolates are grouped into sets according to the matched templates. The next steps are performed in these sets in parallel. The isolate reads are mapped to the template using the mapping algorithm of NDtree[36], yielding equal-length consensus sequences. The Z-score threshold for accepting a base is set to 1.96, and the majority base have to be present in 90% of the mapped reads.

Genetic distance based on nucleotide difference is calculated pairwise between the previous, non-redundant isolates, and the new isolates. Positions with ambiguous bases are discarded. The new isolates are clustered to the non-redundant ones with a threshold of 10, in order to reduce the homology in each set and form informative clusters. In this step, the non-redundant isolate is prioritized over the new isolate and becomes the cluster representative. After the clustering, the remaining new isolates are clustered together with the Hobohm 1 algorithm[43]. In this case, the cluster representative is the one that has already passed the redundancy threshold. The information about new or extended clusters is saved to the main SQLite database. A distance matrix is constructed for all non-redundant isolates and saved to disk for use in the next run. A distance-based phylogenetic tree is inferred by neighbor-joining[44,45]. If there are less than 600 non-redundant isolates in the set, then a whole-genome based approximate maximum likelihood phylogenetic tree is also inferred using IQ-tree[46], where the neighbor-joining tree is the starting tree and the GTR nucleotide substitution model is used. The clustered isolates are placed back onto the clades with zero distances to the cluster representative. Their tip labels start with an asterisk. The information about the trees is saved to the main SQLite database.

When all the phylogenetic trees with new isolates have been inferred, then the main SQLite database is queried for the list of all isolates, their templates, cluster representatives (if there is any) and the latest phylogenetic tree they are on. This information is printed to a tab-separated file.

**Evergreen Online platform**. A query is made to the National Center for Biotechnology Information (NCBI) Sequencing Read Archive (SRA) for the newly published Illumina paired-end sequenced isolates of *Campylobacter jejuni*, *Escherichia coli*, *Listeria monocytogenes*, *Salmonella enterica*, and *Shigella spp*. on a daily basis. Fastq files of raw sequencing reads and the corresponding metadata (collection date, location, institute, source, etc.) are acquired either from SRA or from the European Nucleotide Archive (ENA). The sample inclusion criteria is known metadata for collection date and location, and in addition, samples are included from the following institutions: Unites States Center for Disease Control, United States Food and Drug Administration, Food Safety and Inspection Service, Public Health England, University of Aberdeen, University Hospital Galway, Statens Serum Institut, Norwegian Institute of Public Health. The downloaded isolates are the input to PAPABAC. The metadata are saved in the main SQLite database, and added to the tip labels on the phylogenetic trees.

Individual subtrees are inferred from isolates with less than 20 SNPs distance from each cluster-representative, considering only the positions in the sequences where there is no missing data. No tree is inferred, if no genetic difference is found. The subtrees are inserted into an SQLite database.

Once all instances of the second wrapper script have finished, then the SQLite databases are queried for the list of available phylogenetic trees (the maximum likelihood trees preferred over neighbor-joining ones), changes in the clusters and the list of all isolates in the system, which is then used to update the website.

The phylogenetic trees are interactively visualized on the website (https://cge.cbs.dtu.dk/services/Evergreen/) using the Phylocanvas API (http://phylocanvas.org).

For visualization in external programs, such as Microreact[39], the phylogenetic trees can be downloaded as newick files and the corresponding metadata as tab separated files. The isolates and clusters can be searched by SRA run ID, which allows the quick localization of the clusters that increased in size via their cluster representative.

**Architecture**. The pipeline is written in Python 2.7 and Bash in Unix environment. In addition to the standard Anaconda Python 2.7 packages, it also requires ETE Toolkit v3.0[47] and Joblib v0.11 (https://pythonhosted.org/joblib) packages to be installed. Neighbor program from the PHYLIP package v3.697 (http://evolution.genetics.washington.edu/phylip.html) and IQ-tree v1.6.4[46] are used for the phylogenetic tree inference. The SQL database management is performed with SQLite v3.20.1 (https://www.sqlite.org).

The two main parts of the pipeline have their own wrapper scripts. PAPABAC can be run on a personal computer with as few as four cores. Evergreen Online is running on a high-performance computing cluster, utilizing the Torque (Adaptive Computing Inc., USA) job scheduler. The first wrapper is run in one instance on 20 cores, meanwhile the second wrapper is run once on 20 cores for each template that has at least one new run, in a parallel fashion. When all of these instances are finished running, a Bash script is launched to collect the information from the SQL database, the website is updated and the job for the next day is scheduled.

**Reference database**. The reference sequences are complete prokaryotic chromosomal genomes from the NCBI RefSeq database. Homology reduction was performed at a 99.0% sequence identity threshold with the Hobohm 1 algorithm. The curated NCBI prokaryotic reference genomes were given priority in the process. The reference sequences could be downloaded via ftp (ftp://ftp.cbs.dtu.dk/public/CGE/databases/Evergreen/).

**Computational time comparison**. One hundred and one samples from the Escherichia coli in vitro evolution experiment dataset by Ahrenfeldt et al. were batched according to their sampling time. The parallelization in PAPABAC was disabled. The traditional method meant that the analysis was carried out on all the samples up to the given batch, starting anew each time, but using the same scripts as PAPABAC.

**Benchmarking with dataset by Ahrenfeldt et al**. The last samples in each lineage of the Escherichia coli in vitro evolution experiment dataset[37] were selected for the benchmarking. Therefore, the benchmarking dataset constituted 50 tips on the ideal phylogeny. These samples were batched according to their sampling time (6th, 7th, and 8th day). The batches were processed by PAPABAC chronologically. The pipeline was run with the default parameters. Both maximum likelihood and neighbor-joining trees were inferred.

The phylogenetic trees inferred on all 50 isolates were trimmed for the reference sequence and compared with the ideal phylogeny using the phytools R package (v0.6–60)[48]. The normalized Robinson-Foulds distance was calculated between the ideal and the maximum likelihood, and the ideal and the neighbor-joining trees, after the clustered isolates are removed from each pair of trees. The RF.dist function was utilized from the phangorn R package (v2.4.0)[49].

**Benchmarking with datasets from Timme et al**. The *Campylobacter jejuni* 0810PADBR-1 and the *Listeria monocytogenes* 1408MLGX6-3WGS dataset[38] was downloaded with the provided script into a distinct directory. The pipeline was run individually on the datasets with default parameters. If the isolates were mapped to more than one template, the phylogenetic trees of the template set with the highest number of isolates were evaluated. The maximum likelihood trees were visually compared to the ideal phylogenies and checked for the distribution of the isolates amongst the clades.

Using the default options on the pipeline, the *L. monocytogenes* SNP dataset was added to a copy of the Listeria_monocytogenes_07PF0776_NC_017728_1 template set of Evergreen Online on 2018-06-15.

**Comparison with the NCBI pathogen detection platform**. *Escherichia coli* isolates were queried from the SQL database of Evergreen Online for the period of 2018-03-15 and 2018-06-01, corresponding to a multistate outbreak of *E.coli* O157: H7 in the USA[50]. These samples were subtyped using traditional MLST[2], as it was assumed, that the sequence type with the most isolates would also include the outbreak samples. Sequence type 11, which commonly corresponds to the O157:H7 serotype, was selected for further analysis. The corresponding samples and their SNP clusters were found in the NCBI-PD platform. The phylogenetic tree for the SNP cluster with the most samples (PDS000000952.271) was downloaded. The common samples (Supplementary Data 2) were noted on both the NCBI-PD and the Evergreen Online phylogenetic tree (Escherichia_coli_O157_H7_str_Sakai_chromosome_NC_002695_1, downloaded on 2018-08-07). The refined subtree around SRR6766978 was downloaded on 2019-10-30, and pruned to contain only the isolates that were on the Evergreen Online tree. The common samples on the three biggest clusters on the Evergreen Online tree were labeled, and their placement on the NCBI-PD tree and the refined tree was visually inspected.

**Statistics and reproducibility**. The code for running PAPABAC is provided, and running it on the same input data would yield reproducible results for the consensus sequences, clusters and neighbor-joining trees, for they are produced with deterministic algorithms.

**Reporting summary**. Further information on research design is available in the Nature Research Reporting Summary linked to this article.

## Data availability
No novel datasets were generated during the current study. All analyzed data are available in this published article, or on the website: https://cge.cbs.dtu.dk/services/Evergreen/.

## Code availability
Scripts and installation instructions for the pipeline are publicly available on Bitbucket: https://bitbucket.org/genomicepidemiology/evergreen.

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

## Acknowledgements

The authors would like to thank every public health institute, that releases their sequencing data to the public in a prompt manner. Global surveillance would not be possible without the sharing of data. This project was part of"Collaborative management platform for detection and analyses of (re-) emerging and foodborne outbreaks in Europe" (COMPARE), that has received funding from the European Union's Horizon 2020 research and innovation program under grant agreement No. 643476.

## Author contributions

O. L. developed the overall idea for the project. J.Sz. and J.A. contributed equally: both contributed to the design of the pipeline, wrote the code and wrote the manuscript. J.Sz. carried out the analysis. J.L.B.C. and M.C.F.T. created software that was used in the work. O.L. and F.M.A. advised the study and helped to write the manuscript.

## Competing interests

The authors declare no competing interests.
