## [Peer Review File · Communications Biology]

Reviewers' comments:

Reviewer #1 (Remarks to the Author):

Szarvas et al. present a pipeline, 'PAPABAC' for the analysis of bacterial genomes, which is paired with a visualisation tool, 'Evergreen Online' for displaying the resulting trees. The aim of this approach is to identify outbreaks, in particular of foodborne pathogens. The pipeline is reference-based, and computational efficiency is improved by removing isolates which are less than 10 SNPs from another isolate, inferring the tree using the 'representative' isolates only, and then placing the removed isolates back onto the tree with 0 distance from the representative isolate for that cluster. New publicly available data from the SRA/ENA is downloaded every day, and the trees are re-inferred to incorporate the new data. The code is open-source, the analysis is sound, and the paper is clear and well written.

The main motivation of the paper, a resource of continuously updating trees of bacterial pathogens, is important, timely, and would be of interest to public health epidemiologists in all settings. However, there are several related tools which are completely omitted from the introduction. Nextstrain (<https://doi.org/10.1093/bioinformatics/bty407>) also displays continuously updated trees of certain pathogens. Although it is mainly focussed on viral pathogens, Tuberculosis is listed on the website. Nextstrain offers more functionality than Evergreen Online, such as more tree visualisation options, an interactive map, and a timeline. Microreact (<https://dx.doi.org/10.1099/mgen.0.000093>) offers similar functionality to Nextstrain, but is very easy to upload data to, and has extensive functionality for colouring and labelling isolates by their metadata fields. Enterobase (<https://doi.org/10.1371/journal.pgen.1007261>) continuously downloads publicly available data of foodborne pathogens, but offers far more functionality than Evergreen Online (assemblies, ST typing, sortable metadata). Pathogen.watch (<https://pathogen.watch>) does not continuously download data, but it plots the samples on an interactive map, does ST typing, predicts antibiotic resistance, in addition to displaying an interactive phylogenetic tree. In contrast, the functionality offered by Evergreen Online is extremely limited - namely a phylogenetic tree to aid detection of outbreaks.

Because this is the only functionality offered by Evergreen Online, it should perform this extremely well. However, it is not clear that this is the case. The algorithm for inferring the trees is reasonable for a quick overview of the population, but does not give enough detail. For example it should be possible to zoom in on a cluster, and re-infer a high-resolution subtree of only these isolates - this would not be computationally demanding, and is vital to provide information on outbreak progression. It is also not clear why 99% is used for the initial clustering, as this fragments the species into several sets, each based on a reference genome. This makes it difficult to understand how these trees relate to each other. It would make more sense to have a rougher overall tree for each species, and then be able to zoom in and re-infer a subtree for a potential outbreak cluster (using a member of the cluster as a reference). The rough overall tree could be inferred using mash/fastANI, and this would remove the need to infer the tree using only the reference isolates for each cluster.

The web-based visualisation tool is quite clunky and limited. When all trees are listed, there are many for each species, and it is not possible to know how many and which isolates they contain until the tree is viewed. There are only two tree visualisation layouts, it is not possible to subset or colour by metadata, and there is no map, which is a severe limitation. For example, the main competitors in this field (Microreact and Nextstrain), offers all this functionality and more, and are not cited.

In summary, neither the analysis algorithm or visualisation tool are particularly novel, and both have alternatives with more functionality. However, the ability to search for isolates by SRA ids could be very useful. If the underlying trees (both representative of the whole species, and high-resolution

trees of individual clades) and metadata could be displayed in Microreact in an automated and continuously updating manner, then this would be a powerful tool which would be very useful. Unfortunately, Evergreen Online in its current state is too limited to be widely useful.

Specific comments:

P. 2, L. 42: More detail on what else WGS can do (AMR prediction, detect virulence genes etc) is needed here.

P. 2-3, L. 49-82: Discuss and cite nextstrain, enterobase, pathogen.watch, and microreact

P. 3, Figure 2: the 'ideal' tree on the left has extremely regular branch lengths, compared with ML tree on the right. Why is this? The links between isolates should be shown on this tree, as they are on the supplementary version of the same figure (Figure S2).

P. 6, L. 139: A statistical comparison of the trees should be provided.

P. 6, L. 143: This comparison is more relevant than that with the in-vitro experiment, and so I think Figure S3 should be in the main text.

P. 8, L. 184: How were these isolates selected?

P. 8, L. 191: Why were these trees so different? A high resolution ML tree should be inferred for just this clade, as this should help identify which tree (Evergreen or NCBI) is more accurate.

P. 10, L. 220-230: Why was 10 SNPs chosen as a threshold? How do the results change if this threshold is changed?

P. 11, L. 271: Again, a high resolution ML tree of the clade should be used to resolve this.

Reviewer #2 (Remarks to the Author):

Review:

Title: Large scale automated phylogenomic analysis of bacterial whole-genome isolates and the Evergreen platform

Authors: Judit Szarvas & Johanne Ahrenfeldt, Jose Luis Bellod Cisneros, Martin Christen Frølund Thomsen, Frank M. Aarestrup, Ole Lund

Summary: The authors present a new phylogenomic pipeline and web platform that performs daily clustering of publicly released pathogen genomes - effectively standing up a genomic surveillance system for a targeted set of pathogen species. This is an enormous task and one that nicely supplements previously released tools within the DTU Center for Genomic Epidemiology. The authors present their pipeline approach and present results from appropriate benchmarking datasets. I would recommend a scientific editor for some text smoothing in areas, but overall the writing is clear and well referenced. This new tool is potentially very valuable to the community - there's only one other comparable tool available that has yet to be published. My review focuses on wanting more clarification around advantages of this new tool, cleaning up some inaccuracies in describing the

existing tools, and better targeting the audience so that the reader has a clear understanding of who this tool would benefit most.

Major revisions:

1. After reading this manuscript, I'm left wondering why someone would use the Evergreen platform over NCBI's Pathogen Detection. There seems to be a significant lag for when the raw data is available within Evergreen (e.g. I wasn't able to find any of the data flagged as new within NCBI-PD). I think the authors need to provide more clarity around the advantages of this new pipeline and user interface. Broader set of repositories included? Maybe there is a logical workflow across the larger set of DTU services that makes Evergreen a preferred choice? Is it faster?

2. I think the authors would benefit from more clearly defining the target audience for this tool, then speaking directly to them in this manuscript. Are the target users employees in public health labs? Academic researchers? Epis within Danish public health labs? PulseNet European Epis? COMPARE members? Users of other DTU web services? Once an audience is identified describing a couple use-cases would really help the reader understand its utility. What type of questions are asked and how does Evergreen help answer them? How does evergreen do this better than the NCBI-PD interface?

3. Table 1 needs some correction/updating:

- For a wide range of bacterial species. NCBI provides clustering surveillance for 28 pathogen species. What does "wide range" mean here and why doesn't NCBI-PD meet that threshold?
- Requires only raw sequencing reads as input. Correction: NCBI-PD accepts assemblies and raw reads as input.
- Quality control steps. What are these? Include a description in the methods.
- Communicable nomenclature for subtype and cluster. This appears to be a huge strength of the platform, but as a reader I'm left confused about how Evergreen nomenclature works. Expanding on this feature and contrasting it with other nomenclatures (phylogenetic "addresses" and cgMLST) would really strengthen the manuscript.
- Stable clustering of samples across runs. What does "across runs" mean? Are you making the point that the clusters never change over time? Is this tied to the nomenclature feature above?

4. Would be really nice to have a figure or paragraph outlining dataflow into the Evergreen platform so users understand which isolates they should expect to see in the platform vs. NCBI-PD.

a. Is this correct? Evergreen = all of SRA and ENA. NCBI-PD = SRA data from a list of flagged public-health related bioprojects plus all assembled GenBank genomes.

b. Describe the QC pipeline implemented within Evergreen and list associated thresholds for inclusion in the clustering analysis. QC is listed in table 1, but not described in methods.

c. Authors say that daily updates are done within Evergreen, but none of the new isolates highlighted within NCBI-PD are present in Evergreen. For example, on March 26 the following new Salmonella SRRs are present in NCBI-PD, but not Evergreen: SRR8775497, SRR8775496, SRR8775495. The salmonella pipeline is the lengthiest for NCBI-PD, so I would expect these to show up in the Evergreen platform sooner if their pipeline is indeed faster.

5. Because the backbone of this pipeline is NCBI RefSeq's database, I'm curious what the authors think about the representative nature this database. If someone were to design a perfect reference database, you'd want even coverage spanning the diversity of each species. Is this what you see in the references? Are there holes in the diversity? If so, what are the effects on the pipeline? Is there any evidence of acquisition bias caused by reference genomes being too distant?

Minor revisions:

Lines 50-57. The summary of GenomeTrakr and NCBI's Pathogen detection isn't quite right here.

Consider using the following language.

Since 2012, the US Food and Drug Administration (FDA) has lead a network of public health and university laboratories, called GenomeTrakr. These laboratories sequence bacterial isolates from food and environmental samples and upload raw data to databases at the National Center for Biotechnology Information (NCBI)¹². NCBI's Pathogen Detection portal computes clustering results for 28 pathogenic species, broadly expanding GenomeTrakr's initial foodborne surveillance effort. The NCBI Pathogen Detection pipeline¹³ assembles the samples into draft genomes, scans the current clusters for a within-50-SNP match, then constructs phylogenetic trees for each cluster using an exact maximum compatibility algorithm¹⁴. This approach requires publicly available data from raw or assembled genome submissions and requires extensive computational resources for the larger databases, like Salmonella. No sub-species taxonomical classification has been implemented within this pipeline at this time.

Lines 58-71. This paragraph needs some clarification. Setting the MLST scheme discussion aside, the overall dataflow presented here isn't accurate. This sentence is factually incorrect: "Meanwhile, all of the raw data could be kept locally." This sentence might be true for the European members of PulseNet, but it's not accurate for PulseNet USA: "Only data from individual strains would have to be shared when further confirmation of an outbreak is required". All raw PulseNet USA data are uploaded to the NCBI's public Pathogen Detection Pipeline in real-time, where they are clustered with other public surveillance efforts (GenomeTrakr, PHE, etc.), resulting in browsable SNP trees. PulseNet USA also analyzes their data with 3rd party tools using a suite of cgMLST schemes. The data flow within this 3rd party system is separate from the public facing discussion. I think this distinction is really important – the US is united in publicly releasing all raw data for our foodborne pathogen surveillance. This paragraph makes it seem like there are two separate efforts within the US, which is not the case. Perhaps clarifying US vs. European PulseNet would help here? Or, you can add PN USA to the previous paragraph for the open data effort at NCBI's pathogen detection, then contrast that to the closed system within Europe in the following paragraph.

Lines 72-73. I had to re-read this paragraph a couple times to understand the point. I think the authors are saying that analysts at public health labs often view the SNP trees at NCBI and dendograms within BioNumerics as preliminary results. That these results sometimes need further refinement using in-house tools? Perhaps narrowing or expanding taxon sampling, etc. Maybe more clarity in the first sentence would set up the rest of the paragraph? For the most part this further refinement has now become rare in the US as our existing "first pass" tools have become quite good. If this is what the authors meant, how does Evergreen offer this further refinement?

Lines 83-90. Making a clearer contrast to NCBI's Pathogen Detection portal and BioNumerics cgMLST would be good to highlight Evergreen's advantages in addition to a stable naming scheme. For example, what are some of the known limitations of current gene- and SNP-based approaches that Evergreen overcomes?

Lines 152-154: "Raw WGS data files of five major foodborne pathogens (*C. jejuni*, *E. coli*, *L. monocytogenes*, *Salmonella enterica*, and *Shigella* spp.) are downloaded daily from public repositories . . .". Please list the public repositories that feed Evergreen its data. For example, is it scanning all of SRA and ENA? Or just a targeted set of BioProjects and/or submitters? How up-to-date is evergreen in comparison to NCBI-PD? If a set of SRRs found in NCBI-PD are not available in Evergreen is this because there's a lag in updating the system? Or they didn't pass Evergreen QC? Or evergreen monitors a narrower set of repositories than does NCBI-PD?

Lines 487-490. Figure S3. Please include a description of what the asterisks mean. It appears the PAPABAC pipeline is recovering fewer SNPs, resulting in lower resolution trees within these benchmark

datasets (e.g. for the Lm tree NCBI-PD finds about 13 SNP differences between the two major blue OB lineages, but PAPABAC appears to find next to zero based on branch lengths). Is this as expected? Please comment on this observation in the discussion.

Editorial notes:

Title: "phylogenomical". Does this have a different meaning than commonly used word "phylogenomic"? I've never seen this word before and it doesn't turn up any google hits. I would recommend sticking to commonly used terminology unless you require something new, in which case I would recommend defining this new word upfront in the Introduction.

Line 50: GenomeTrackr should be spelled "GenomeTrakr" throughout the manuscript.

Line 50-51. "bacterial isolates from clinical and environmental samples..." Should read "...bacterial isolates from food and environmental samples..."

Line 303: update link for GenomeTrakr:

<https://www.fda.gov/Food/FoodScienceResearch/WholeGenomeSequencingProgramWGS/default.htm>

Line 309. Curious what this reference is? This recent paper I saw might be a better reference: Timme RE, Sanchez Leon M, Allard MW 2019. Utilizing the Public GenomeTrakr Database for Foodborne Pathogen Traceback. *Methods in molecular biology* (Clifton, N.J.) 1918:201–212. DOI: 10.1007/978-1-4939-9000-9_17.

Reviewer #3 (Remarks to the Author):

Dear authors,

This is a well written manuscript that proposes a solution to the rapidly building volume of WGS surveillance data worldwide that are accumulating in silo. The authors have created a bioinformatics pipeline that allows rapid and continuous phylogenomic analysis of large scale genomics data.

This tool is of great value and addresses the gaps in existing databases and tools such as NCBI-PD and Snapper DB. These include allowing for raw sequencing reads as input, being assembly-free, has built in automated phylogenomic analysis, and are open source.

The authors have also addressed the issue of tool validation of the PAPABAC pipeline, and the authors clearly appreciate the importance of cluster generation both locally and across different databases. However, it is not clear whether the raw data can be downloadable for further analysis into the user's server, or whether analysis and comparisons can only be performed on Evergreen. This would be important to clarify.

While I have tried hard to find flaws in this manuscript, I am unable to, and am thrilled to see this tool readily available for WGS surveillance work internationally. I look forward to this tool being a mainstream means for communicating public health related clusters across international boundaries.

I support the publication of this manuscript with minor edits as outlined above.

Dear Reviewers,

Thank you for taking time to review and comment on our manuscript. We have revised the manuscript in line with your comments and provided answers to the raised questions below.

Reviewer #1 highlighted the importance of the visualization of the results, therefore we implemented external visualisation with Microreact in addition to the regular export of the phylogenetic trees.

Reviewer #2 mostly compared our method with NCBI-PD, a method which is neither published nor available as software. This makes any benchmarking against NCBI-PD quite difficult, but nevertheless we have put effort into doing this in the best way possible under these circumstances. The reviewer asks who will be interested in our method. To this our answer is that many will be interested in running the software presented here, which have been documented and validated via a scientific publication. The source code is publicly available, therefore users can run it on their own computers on their own data, and it will aid in the bias-free discovery of more outbreak clusters.

Best regards,

Szarvas et al

Our comments:

Referee expertise:

Referee #1: Pathogenomics

Referee #2: Phylogenomic pipelines for pathogen surveillance

Referee #3: Molecular detection of bacterial pathogens

Reviewers' comments:

Reviewer #1 (Remarks to the Author):

Szarvas et al. present a pipeline, 'PAPABAC' for the analysis of bacterial genomes, which is paired with a visualisation tool, 'Evergreen Online' for displaying the resulting trees. The aim of this approach is to identify outbreaks, in particular of foodborne pathogens. The pipeline is reference-based, and computational efficiency is improved by removing isolates which are less than 10 SNPs from another isolate, inferring the tree using the 'representative' isolates only, and then placing the removed isolates back onto the tree with 0 distance from the representative isolate for that cluster. New publicly available data from the SRA/ENA is downloaded every day, and the trees are re-inferred to incorporate the new data. The code is open-source, the analysis is sound, and the paper is clear and well written.

The main motivation of the paper, a resource of continuously updating trees of bacterial pathogens, is important, timely, and would be of interest to public health epidemiologists in all settings. However, there are several related tools which are completely omitted from the introduction. Nextstrain (<https://doi.org/10.1093/bioinformatics/bty407>) also displays continuously updated trees of certain

pathogens. Although it is mainly focussed on viral pathogens, Tuberculosis is listed on the website. Nextstrain offers more functionality than Evergreen Online, such as more tree visualisation options, an interactive map, and a timeline.

Nextstrain is, as the reviewer notes, focused on viruses, and the 999 Tuberculosis strains covered only goes until August 2018, and is thus not continuously updated. Moreover, its bioinformatics pipeline Augur takes in consensus sequences, not raw sequencing data, and their workflow differs from PAPABAC. We agree that the Nextstrain interface has more interactive features. At the moment NextStrain covers only 9 species, and the largest tree contains 2,255 samples. PAPABAC is designed to handle much larger datasets (160,000 isolates so far). It is unknown if Nextstrain will be able to handle such a large dataset, and if it can, whether the visualizations will be usable for the user. [Comment only]

Microreact (<https://dx.doi.org/10.1099/mgen.0.000093>) offers similar functionality to Nextstrain, but is very easy to upload data to, and has extensive functionality for colouring and labelling isolates by their metadata fields.

We agree again that Microreact has a nice user interface, but the completely comment misses the point that Evergreen Online is developed to handle the analysis and visualization of datasets that are two orders of magnitude larger. And Microreact is not analyzing raw NGS data, but simply visualizing outputs. We have added support to visualize the Evergreen trees via Microreact (<https://microreact.org/upload>) by making the required .tsv and .nwk files available at a publicly available URL under <https://cge.cbs.dtu.dk/services/Evergreen> and via their API. [Modified code, Methods L446-448]

Enterobase (<https://doi.org/10.1371/journal.pgen.1007261>) continuously downloads publicly available data of foodborne pathogens, but offers far more functionality than Evergreen Online (assemblies, ST typing, sortable metadata).

Enterobase has been cited and the pros and cons of using cgMLST rather than SNPs for phylogeny is extensively discussed. In brief, we believe one of the limitations with cgMLST is the need for a globally agreed and updated allele database for all species to be covered. Furthermore, focusing on the core genome may miss some mutations that are important to track recent evolution. Enterobase is only available for a few bacterial species, whereas Evergreen can cover all bacteria (and potential other) species. [Introduction L54-70]

Pathogen.watch (<https://pathogen.watch>) does not continuously download data, but it plots the samples on an interactive map, does ST typing, predicts antibiotic resistance, in addition to displaying an interactive phylogenetic tree. In contrast, the functionality offered by Evergreen Online is extremely limited - namely a phylogenetic tree to aid detection of outbreaks.

As the reviewer notes pathogen.watch does NOT continuously update with new genomes and this is a very important difference with Evergreen Online. The main accomplishment of the PAPABAC method is that it can update SNP trees with 160,000+ isolates in real time and that we have a hard time seeing as an “extremely limited functionality”. Many of the other phenotype-predictions are already available from our web site cge.cbs.dtu.dk. In the future we may combine these with Evergreen Online, but it is beyond the scope of the current paper. [Introduction L68-69]

Because this is the only functionality offered by Evergreen Online, it should perform this extremely well. However, it is not clear that this is the case. The algorithm for inferring the trees is reasonable for a quick overview of the population, but does not give enough detail. For example it should be possible to zoom in on a cluster, and re-infer a high-resolution subtree of only these isolates - this would not be computationally demanding, and is vital to provide information on outbreak progression.

We have now added the calculation of a high resolution tree for each cluster with non-zero diversity in Evergreen Online. [Modified code, Methods L441-443]

It is also not clear why 99% is used for the initial clustering, as this fragments the species into several sets, each based on a reference genome. This makes it difficult to understand how these trees relate to each other. It would make more sense to have a rougher overall tree for each species, and then be able to zoom in and re-infer a subtree for a potential outbreak cluster (using a member of the cluster as a reference). The rough overall tree could be inferred using mash/fastANI, and this would remove the need to infer the tree using only the reference isolates for each cluster.

We initially tried to find a %ID to correspond to serotypes but found no good correlation. The 99% cluster means for example in E.coli, that isolates will be in the same tree if they have less than approx. 1% times 5 million bases = 50.000 SNPs, meaning that if they are not in the same tree they are definitely not in the same outbreak. We are not convinced that merging all trees into one tree would make it easier for the user to overview the data. We think our solution of making a table shortlisting only those trees where clusters are appearing or expanding will make it easier for epidemiologist to focus on the important areas. [Comment only]

The web-based visualisation tool is quite clunky and limited. When all trees are listed, there are many for each species,

Yes, there are several trees per species, but given that some of these contains thousands of isolates, we think this is preferable to have one big tree. Even with the number of isolates present in some of the trees, Phylocanvas, which we use to visualize trees becomes very slow, which is why we as default only show a tree with six levels of branching from the identified outbreak. [Comment only]

and it is not possible to know how many and which isolates they contain until the tree is viewed.

We have added the information on the number of samples to the website. [Modified code]

There are only two tree visualisation layouts, it is not possible to subset or colour by metadata, and there is no map, which is a severe limitation. For example, the main competitors in this field (Microreact and Nextstrain), offers all this functionality and more, and are not cited.

Again, we do not see Microreact and Nextstrain as competitors in this field, but rather as very nice visualization tools that would complement PAPABAC very nicely. We have added this to the paper, and we have added a functionality to export data to Microreact. [Modified code, Methods L446-448]

In summary, neither the analysis algorithm or visualisation tool are particularly novel, and both have alternatives with more functionality.

This statement is clearly wrong. None of the cited methods have documented ability to classify 160.000+ strains in SNP trees. The only one who comes close is Enterobase which have published that their methods can handle 100.000+ isolates, but that is with cgMLST and not with SNPs. In addition, Evergreen Online can be updated in real-time, which is a feature that, to our knowledge, no other published method has. [Comment only]

However, the ability to search for isolates by SRA ids could be very useful.

It is possible to search for SRA ids. [Comment only]

If the underlying trees (both representative of the whole species, and high-resolution trees of individual clades) and metadata could be displayed in Microreact in an automated and continuously updating manner, then this would be a powerful tool which would be very useful. Unfortunately, Evergreen Online in its current state is too limited to be widely useful.

The newick files are already available. As noted above we have now added a functionality to export metadata to Microreact in the form of a csv file. [Modified code, Methods L446-448]

Specific comments:

P. 2, L. 42: More detail on what else WGS can do (AMR prediction, detect virulence genes etc) is needed here.

This isn't the focus of this manuscript. [Comment only]

P. 2-3, L. 49-82: Discuss and cite nextstrain, enterobase, pathogen.watch, and microreact

We cited Enterobase, pathogen.watch and Microreact. As we argued before, Nextstrain is primarily for the visualization of viral phylodynamic analysis results, therefore we don't see the relevance of citing it in our manuscript about a bacterial phylogenomic analysis pipeline.

P. 3, Figure 2: the 'ideal' tree on the left has extremely regular branch lengths, compared with ML tree on the right. Why is this? The links between isolates should be shown on this tree, as they are on the supplementary version of the same figure (Figure S2).

This is because it is a tree artificially constructed as it would be if the mutation rate is constant. Links are now displayed in figure. [Updated Figure 2]

P. 6, L. 139: A statistical comparison of the trees should be provided.

This has been added to the Results section, L143-145.

P. 6, L. 143: This comparison is more relevant than that with the in-vitro experiment, and so I think Figure S3 should be in the main text.

We believe, that the sensitivity of the method is better illustrated with this figure, as the SNP distances between the outbreak, non-outbreak samples are higher in Fig. S3. [Comment only]

P. 8, L. 184: How were these isolates selected?

This is described in the Methods section, L493-502.

P. 8, L. 191: Why were these trees so different? A high resolution ML tree should be inferred for just this clade, as this should help identify which tree (Evergreen or NCBI) is more accurate.

The root cause of the difference between the two trees is the clustering step at 10 SNPs that PAPABAC makes, and the low level genetic diversity in the selected clusters. This is covered in detail in the Discussion, L270-276.

P. 10, L. 220-230: Why was 10 SNPs chosen as a threshold? How do the results change if this threshold is changed?

10 SNP was chosen because it is the generally used threshold for being sure that two isolates are from the same outbreak. Increasing the threshold results in the expansion and sometimes merging of the original clusters and the emergence of new clusters. [Comment only]

P. 11, L. 271: Again, a high resolution ML tree of the clade should be used to resolve this.

See above.

Reviewer #2 (Remarks to the Author):

Review:

Title: Large scale automated phylogenomic analysis of bacterial whole-genome isolates and the Evergreen platform

Authors: Judit Szarvas & Johanne Ahrenfeldt, Jose Luis Bellod Cisneros, Martin Christen Frølund Thomsen, Frank M. Aarestrup, Ole Lund

Summary: The authors present a new phylogenomic pipeline and web platform that performs daily clustering of publicly released pathogen genomes - effectively standing up a genomic surveillance system for a targeted set of pathogen species. This is an enormous task and one that nicely supplements previously released tools within the DTU Center for Genomic Epidemiology. The authors present their pipeline approach and present results from appropriate benchmarking datasets. I would recommend a scientific editor for some text smoothing in areas, but overall the writing is clear and well referenced. This new tool is potentially very valuable to the community - there's only one other comparable tool available that has yet to be published. My review focuses on wanting more clarification around advantages of this new tool, cleaning up some inaccuracies in describing the existing tools, and better targeting the audience so that the reader has a clear understanding of who this tool would benefit most.

Major revisions:

- 1. After reading this manuscript, I'm left wondering why someone would use the Evergreen platform over NCBI's Pathogen Detection. There seems to be a significant lag for when the raw data is available within Evergreen (e.g. I wasn't able to find any of the data flagged as new within NCBI-PD). I think the authors need to provide more clarity around the advantages of this new pipeline and user interface. Broader set of repositories included? Maybe there is a logical workflow across the larger set of DTU services that makes Evergreen a preferred choice? Is it faster?*
- 2. I think the authors would benefit from more clearly defining the target audience for this tool, then speaking directly to them in this manuscript. Are the target users employees in public health labs? Academic researchers? Epis within Danish public health labs? PulseNet European Epis? COMPARE members? Users of other DTU web services? Once an audience is identified describing a couple use-cases would really help the reader understand its utility. What type of questions are asked and how does Evergreen help answer them? How does evergreen do this better than the NCBI-PD interface?*

The NCBI-PD is an unpublished method, and neither is the software publicly available, and thus its accuracy or scientific merit cannot be validated at this time. Being a non NCBI site our speed is limited by the lag before NCBI makes their data available, and the time it takes to reformat the raw reads from NCBI format to the more commonly used fastq format. That said we have still seen examples such as SRR7280611 which were available in Evergreen Online on June 11, 2018, but not at NCBI-PD before June 13 2018.

In brief, PAPABAC is for anyone who wants/needs open standards, open software, validated methods and possibility of local installation. This is the case for many public health institutions where there may be regulatory requirements for transparency of the methods used. The NCBI-PD do not at the moment fulfill these criteria, but we look forward to see this happening sometime in the future as the reviewer hints to.

It fulfills a need that many employees in public health labs have put to us. Every week new isolates are being sequenced, and they need to know if any of the new isolates are similar to each other or to anything they have ever sequenced before. NCBI-PD may not be a feasible tool for this since many hospitals and public health labs are not allowed to share their data in real time, but they could use the PAPABAC pipeline on their own data to discover local outbreaks.

The main target is in public health labs which sequence more in a continuous mode, rather than scientists that often focus on sequencing a batch of isolates related to a scientific study, but the distinction is blurred for example for scientists doing research in new tools for public health. [Comment only]

3. Table 1 needs some correction/updating:

- *For a wide range of bacterial species. NCBI provides clustering surveillance for 28 pathogen species. What does “wide range” mean here and why doesn’t NCBI-PD meet that threshold?*

We were unaware that NCBI have recently expanded to 28 species and have changed the table text to acknowledge this. [Table 1]

- *Requires only raw sequencing reads as input. Correction: NCBI-PD accepts assemblies and raw reads as input.*

We have meant that no additional information or specific configuration was needed in order to run the pipeline. We have amended the text to reflect this. [Table 1]

- *Quality control steps. What are these? Include a description in the methods.*

In short, accepting mapping results higher than 99.0 identity% and a minimum average depth ensures that low quality sequencing runs don’t progress further in the pipeline. Coupled that with the significance threshold for the consensus base calling, the consensus sequences are good quality. We have added a description of this to the Method section, L404-412.

- *Communicable nomenclature for subtype and cluster. This appears to be a huge strength of the platform, but as a reader I’m left confused about how Evergreen nomenclature works. Expanding on this feature and contrasting it with other nomenclatures (phylogenetic “addresses” and cgMLST) would really strengthen the manuscript.*

We have expanded the section explaining this. In brief the combination of the tree (template) and the cluster (representative) uniquely identifies a cluster of isolates (which may contain one or more isolates) [Discussion, L237-240]

- *Stable clustering of samples across runs. What does “across runs” mean? Are you making the point that the clusters never change over time? Is this tied to the nomenclature feature above?*

It means, that once a nomenclature template+representative has been assigned to an isolate, it won’t change even if more isolates are added to the tree in subsequent runs. Clusters don’t merge, or break up. [Comment only]

4. Would be really nice to have a figure or paragraph outlining dataflow into the Evergreen platform so users understand which isolates they should expect to see in the platform vs. NCBI-PD.

- a. Is this correct? Evergreen = all of SRA and ENA. NCBI-PD = SRA data from a list of flagged public-health related bioprojects plus all assembled GenBank genomes.*

We have added a description of which isolates have been included, as we think this is better explained in a text than in a figure. [Methods, Online Evergreen platform, L435-439]

- b. Describe the QC pipeline implemented within Evergreen and list associated thresholds for inclusion in the clustering analysis. QC is listed in table 1, but not described in methods.*

We have expanded the description of our QC in Methods, L404-412

- c. Authors say that daily updates are done within Evergreen, but none of the new isolates highlighted within*

NCBI-PD are present in Evergreen. For example, on March 26 the following new Salmonella SRRs are present in NCBI-PD, but not Evergreen: SRR8775497, SRR8775496, SRR8775495. The salmonella pipeline is the lengthiest for NCBI-PD, so I would expect these to show up in the Evergreen platform sooner if their pipeline is indeed faster.

Our pipeline has always been able to complete the calculations within 24 hours but sometimes the time to download the new isolates have delayed the update of the trees. [Comment only]

5. Because the backbone of this pipeline is NCBI RefSeq's database, I'm curious what the authors think about the representative nature this database. If someone were to design a perfect reference database, you'd want even coverage spanning the diversity of each species. Is this what you see in the references?

We do not show templates that are more than 99% similar, so this evens out the case where there are many very similar templates in the database. [Comment only]

Are there holes in the diversity?

Indeed, there are sequence spaces that are not spanned by complete genomes, and sequencing efforts in these areas should be encouraged. Currently, if a close template is not found for an isolate it is not included. In our experience, clinically important references genomes generally provide sufficient diversity to cover the sequence space of interest. [Comment only]

If so, what are the effects on the pipeline? Is there any evidence of acquisition bias caused by reference genomes being too distant?

New templates can be added to the database to accommodate this. In principle, each isolate that was not similar to any template can be added to the list of templates, and this will not change any existing trees or nomenclature assignments. [Comment only]

Minor revisions:

Lines 50-57. The summary of GenomeTrakr and NCBI's Pathogen detection isn't quite right here. Consider using the following language.

Since 2012, the US Food and Drug Administration (FDA) has lead a network of public health and university laboratories, called GenomeTrakr. These laboratories sequence bacterial isolates from food and environmental samples and upload raw data to databases at the National Center for Biotechnology Information (NCBI)¹². NCBI's Pathogen Detection portal computes clustering results for 28 pathogenic species, broadly expanding GenomeTrakr's initial foodborne surveillance effort. The NCBI Pathogen Detection pipeline¹³ assembles the samples into draft genomes, scans the current clusters for a within-50-SNP match, then constructs phylogenetic trees for each cluster using an exact maximum compatibility algorithm¹⁴. This approach requires publicly available data from raw or assembled genome submissions and requires extensive computational resources for the larger databases, like Salmonella. No sub-species taxonomical classification has been implemented within this pipeline at this time.

We have edited the paragraph to reflect these facts. [Introduction, L44-53]

Lines 58-71. This paragraph needs some clarification. Setting the MLST scheme discussion aside, the overall dataflow presented here isn't accurate. This sentence is factually incorrect: "Meanwhile, all of the raw data could be kept locally." This sentence might be true for the European members of PulseNet, but it's not accurate for PulseNet USA: "Only data from individual strains would have to be shared when further confirmation of an outbreak is required". All raw PulseNet USA data are uploaded to the NCBI's public Pathogen Detection Pipeline in real-time, where they are clustered with other public surveillance efforts (GenomeTrakr, PHE, etc.), resulting in browsable SNP trees. PulseNet USA also analyzes their data with 3rd party tools using a suite of cgMLST schemes. The data flow within this 3rd party system is separate from the public facing discussion. I think this distinction is really important – the US is united in publicly releasing all raw data

for our foodborne pathogen surveillance. This paragraph makes it seem like there are two separate efforts within the US, which is not the case. Perhaps clarifying US vs. European PulseNet would help here? Or, you can add PN USA to the previous paragraph for the open data effort at NCBI's pathogen detection, then contrast that to the closed system within Europe in the following paragraph.

We meant that it was technically possible to keep data locally, but there may of course be legal requirements to share data. We have revised the text to clarify this. [Introduction, L57-59]

Lines 72-73. I had to re-read this paragraph a couple times to understand the point. I think the authors are saying that analysts at public health labs often view the SNP trees at NCBI and dendograms within BioNumerics as preliminary results. That these results sometimes need further refinement using in-house tools? Perhaps narrowing or expanding taxon sampling, etc. Maybe more clarity in the first sentence would set up the rest of the paragraph? For the most part this further refinement has now become rare in the US as our existing "first pass" tools have become quite good. If this is what the authors meant, how does Evergreen offer this further refinement?

We meant that usually, phylogenetic analysis is done on selected samples, that were assembled, subtyped, phenotyped beforehand. With PAPABAC, phylogenetic analysis can be the first step, as no pre-knowledge about the input data is necessary. Afterwards, the further refinement with in-house methods can be carried out on the clusters that PAPABAC found. We believe, that by performing a bias-free clustering first, more clusters could be detected. [Introduction, L71-73]

Lines 83-90. Making a clearer contrast to NCBI's Pathogen Detection portal and BioNumerics cgMLST would be good to highlight Evergreen's advantages in addition to a stable naming scheme. For example, what are some of the known limitations of current gene- and SNP-based approaches that Evergreen overcomes?

As the introduction section states, selection and curation of cg- and wg-MLST schemes is a limitation to gene-based approaches. Moreover, evolution is ongoing, and calling alleles is not always straightforward process. Reference based SNP-approaches overcome these limitations, but choosing the right reference is crucial. Moreover, with the increasing number of isolates in surveillance, clusters could be overlooked by not including all the isolates involved in an outbreak in a phylogenetic analysis. Our pipeline solves this problem by automatization of the reference selection and isolate-classification based on rapid subtyping. [Comment only]

Lines 152-154: "Raw WGS data files of five major foodborne pathogens (C. jejuni, E. 152 coli, L. monocytogenes, Salmonella enterica, and Shigella spp.) are downloaded daily from public repositories . . .". Please list the public repositories that feed Evergreen its data. For example, is it scanning all of SRA and ENA? Or just a targeted set of BioProjects and/or submitters? How up-to-date is evergreen in comparison to NCBI-PD? If a set of SRRs found in NCBI-PD are not available in Evergreen is this because there's a lag in updating the system? Or they didn't pass Evergreen QC? Or evergreen monitors a narrower set of repositories than does NCBI-PD?

More details on the inclusion criteria for Evergreen Online have been added, as noted before. [Methods, Online Evergreen platform, L435-439]

Lines 487-490. Figure S3. Please include a description of what the asterisks mean. It appears the PAPABAC pipeline is recovering fewer SNPs, resulting in lower resolution trees within these benchmark datasets (e.g. for the Lm tree NCBI-PD finds about 13 SNP differences between the two major blue OB lineages, but PAPABAC appears to find next to zero based on branch lengths). Is this as expected? Please comment on this observation in the discussion.

Have added the explanation for the asterisk to each figure label. The explanation for the shorter branch lengths, in brief, is that the trees on the left were inferred using the CFSAN SNP pipeline (<https://doi.org/10.7717/peerj-cs.20>), which does extensive filtering on the found SNPs, and collects them into a SNP matrix, that is then the input for the ML method. Therefore it has a higher value for branch

length, than PAPABAC does. The zero length branch length inside a cluster is a deliberate choice, as no distances are calculated between the isolated in the cluster. And as the cluster representative was within 10 SNPs of the isolates of the two major lineages, they were merged into one cluster.

Editorial notes:

Title: “phylogenomical”. Does this have a different meaning than commonly used word “phylogenomic”? I’ve never seen this word before and it doesn’t turn up any google hits. I would recommend sticking to commonly used terminology unless you require something new, in which case I would recommend defining this new word upfront in the Introduction.

We have changed it to phylogenomic. [Title]

Line 50: GenomeTrackr should be spelled “GenomeTrakr” throughout the manuscript.

We have corrected this. [Introduction]

Line 50-51. “bacterial isolates from clinical and environmental samples...” Should read “...bacterial isolates from food and environmental samples...”

This has been corrected. [Introduction, L45]

Line 303: update link for GenomeTrakr:

<https://www.fda.gov/Food/FoodScienceResearch/WholeGenomeSequencingProgramWGS/default.htm>

Have updated the link. [L307]

Line 309. Curious what this reference is? This recent paper I saw might be a better reference:

Timme RE, Sanchez Leon M, Allard MW 2019. Utilizing the Public GenomeTrakr Database for Foodborne Pathogen Traceback. Methods in molecular biology (Clifton, N.J.) 1918:201–212. DOI: 10.1007/978-1-4939-9000-9_17.

We have changed to cite this reference [L313]

Reviewer #3 (Remarks to the Author):

Dear authors,

This is a well written manuscript that proposes a solution to the rapidly building volume of WGS surveillance data worldwide that are accumulating in silo. The authors have created a bioinformatics pipeline that allows rapid and continuous phylogenomic analysis of large scale genomics data.

This tool is of great value and addresses the gaps in existing databases and tools such as NCBI-PD and Snapper DB. These include allowing for raw sequencing reads as input, being assembly-free, has built in automated phylogenomical analysis, and are open source.

The authors have also addressed the issue of tool validation of the PAPABAC pipeline, and the authors clearly appreciate the importance of cluster generation both locally and across different databases. However, it is not clear whether the raw data can be downloadable for further analysis into the user’s server, or whether analysis and comparisons can only be performed on Evergreen. This would be important to clarify.

While I have tried hard to find flaws in this manuscript, I am unable to, and am thrilled to see this tool readily available for WGS surveillance work internationally. I look forward to this tool being a mainstream means for communicating public health related clusters across international boundaries.

I support the publication of this manuscript with minor edits as outlined above.

Dear Reviewers,

Thank you for taking time to review and comment on our manuscript. We have revised the manuscript in line with your comments and provided answers to the raised questions below.

Reviewer #1 highlighted the importance of the visualization of the results, therefore we implemented external visualisation with Microreact in addition to the regular export of the phylogenetic trees.

Reviewer #2 mostly compared our method with NCBI-PD, a method which is neither published nor available as software. This makes any benchmarking against NCBI-PD quite difficult, but nevertheless we have put effort into doing this in the best way possible under these circumstances. The reviewer asks who will be interested in our method. To this our answer is that many will be interested in running the software presented here, which have been documented and validated via a scientific publication. The source code is publicly available, therefore users can run it on their own computers on their own data, and it will aid in the bias-free discovery of more outbreak clusters.

Best regards,

Szarvas et al

Our comments:

Referee expertise:

Referee #1: Pathogenomics

Referee #2: Phylogenomic pipelines for pathogen surveillance

Referee #3: Molecular detection of bacterial pathogens

Reviewers' comments:

Reviewer #1 (Remarks to the Author):

Szarvas et al. present a pipeline, 'PAPABAC' for the analysis of bacterial genomes, which is paired with a visualisation tool, 'Evergreen Online' for displaying the resulting trees. The aim of this approach is to identify outbreaks, in particular of foodborne pathogens. The pipeline is reference-based, and computational efficiency is improved by removing isolates which are less than 10 SNPs from another isolate, inferring the tree using the 'representative' isolates only, and then placing the removed isolates back onto the tree with 0 distance from the representative isolate for that cluster. New publicly available data from the SRA/ENA is downloaded every day, and the trees are re-inferred to incorporate the new data. The code is open-source, the analysis is sound, and the paper is clear and well written.

The main motivation of the paper, a resource of continuously updating trees of bacterial pathogens, is important, timely, and would be of interest to public health epidemiologists in all settings. However, there are several related tools which are completely omitted from the introduction. Nextstrain (<https://doi.org/10.1093/bioinformatics/bty407>) also displays continuously updated trees of certain

pathogens. Although it is mainly focussed on viral pathogens, Tuberculosis is listed on the website. Nextstrain offers more functionality than Evergreen Online, such as more tree visualisation options, an interactive map, and a timeline.

Nextstrain is, as the reviewer notes, focused on viruses, and the 999 Tuberculosis strains covered only goes until August 2018, and is thus not continuously updated. Moreover, its bioinformatics pipeline Augur takes in consensus sequences, not raw sequencing data, and their workflow differs from PAPABAC. We agree that the Nextstrain interface has more interactive features. At the moment NextStrain covers only 9 species, and the largest tree contains 2,255 samples. PAPABAC is designed to handle much larger datasets (160,000 isolates so far). It is unknown if Nextstrain will be able to handle such a large dataset, and if it can, whether the visualizations will be usable for the user. [Comment only]

Regardless of the merits of both approaches, the overlap in scope of the tools means that nextstrain should at least be mentioned.

Microreact (<https://dx.doi.org/10.1099/mgen.0.000093>) offers similar functionality to Nextstrain, but is very easy to upload data to, and has extensive functionality for colouring and labelling isolates by their metadata fields.

We agree again that Microreact has a nice user interface, but the completely comment misses the point that Evergreen Online is developed to handle the analysis and visualization of datasets that are two orders of magnitude larger. And Microreact is not analyzing raw NGS data, but simply visualizing outputs. We have added support to visualize the Evergreen trees via Microreact (<https://microreact.org/upload>) by making the required .tsv and .nwk files available at a publicly available URL under <https://cge.cbs.dtu.dk/services/Evergreen> and via their API. [Modified code, Methods L446-448]

The addition of support for visualisation in microreact is very welcome. Clearly Evergreen is capable of analysing extremely large datasets, however the visualisation tool still struggles with the large trees. For example, the largest tree, “Salmonella_enterica_subsp_enterica_serovar_Typhimurium_str_LT2_NC_003197_2” opens in evergreen, but any attempt to zoom in or click on nodes crashes the browser (on my laptop at least). In contrast, it is possible to interactively view the same tree in microreact (albeit with some lag).

Enterobase (<https://doi.org/10.1371/journal.pgen.1007261>) continuously downloads publicly available data of foodborne pathogens, but offers far more functionality than Evergreen Online (assemblies, ST typing, sortable metadata).

Enterobase has been cited and the pros and cons of using cgMLST rather than SNPs for phylogeny is extensively discussed. In brief, we believe one of the limitations with cgMLST is the need for a globally agreed and updated allele database for all species to be covered. Furthermore, focusing on the core genome may miss some mutations that are important to track recent evolution. Enterobase is only available for a few bacterial species, whereas Evergreen can cover all bacteria (and potential other) species. [Introduction L54-70]

Agreed about the need for a globally agreed and updated allele database, and focusing on the core means some mutations are missed. However, because the PAPABAC approach is reference based, only SNPs in regions present in the reference will be called, and so some recent mutations may still be missed. The extent to which this happens will also vary based on the distance to the reference.

Pathogen.watch (<https://pathogen.watch>) does not continuously download data, but it plots the samples on an interactive map, does ST typing, predicts antibiotic resistance, in addition to displaying an interactive

phylogenetic tree. In contrast, the functionality offered by Evergreen Online is extremely limited - namely a phylogenetic tree to aid detection of outbreaks.

As the reviewer notes pathogen.watch does NOT continuously update with new genomes and this is a very important difference with Evergreen Online. The main accomplishment of the PAPABAC method is that it can update SNP trees with 160.000+ isolates in real time and that we have a hard time seeing as an “extremely limited functionality”. Many of the other phenotype-predictions are already available from our web site cge.cbs.dtu.dk. In the future we may combine these with Evergreen Online, but it is beyond the scope of the current paper. [Introduction L68-69]

Fair enough, the ability to update such large datasets in real time is impressive.

Because this is the only functionality offered by Evergreen Online, it should perform this extremely well. However, it is not clear that this is the case. The algorithm for inferring the trees is reasonable for a quick overview of the population, but does not give enough detail. For example it should be possible to zoom in on a cluster, and re-infer a high-resolution subtree of only these isolates - this would not be computationally demanding, and is vital to provide information on outbreak progression.

We have now added the calculation of a high resolution tree for each cluster with non-zero diversity in Evergreen Online. [Modified code, Methods L441-443]

Where are these trees with non-zero distances? I can only see the original trees with zero distance clusters indicated by asterisks.

It is also not clear why 99% is used for the initial clustering, as this fragments the species into several sets, each based on a reference genome. This makes it difficult to understand how these trees relate to each other. It would make more sense to have a rougher overall tree for each species, and then be able to zoom in and re-infer a subtree for a potential outbreak cluster (using a member of the cluster as a reference). The rough overall tree could be inferred using mash/fastANI, and this would remove the need to infer the tree using only the reference isolates for each cluster.

We initially tried to find a %ID to correspond to serotypes but found no good correlation. The 99% cluster means for example in E.coli, that isolates will be in the same tree if they have less than approx. 1% times 5 million bases = 50.000 SNPs, meaning that if they are not in the same tree they are definitely not in the same outbreak. We are not convinced that merging all trees into one tree would make it easier for the user to overview the data. We think our solution of making a table shortlisting only those trees where clusters are appearing or expanding will make it easier for epidemiologist to focus on the important areas.

[Comment only]

Agreed about a table shortlisting expanding clusters being the easiest way to flag outbreaks.

However, this is not incompatible with a single overview tree per species, and I respectfully disagree that the current approach is more intuitive.

The web-based visualisation tool is quite clunky and limited. When all trees are listed, there are many for each species,

Yes, there are several trees per species, but given that some of these contains thousands of isolates, we think this is preferable to have one big tree. Even with the number of isolates present in some of the trees, PhyloCanvas, which we use to visualize trees becomes very slow, which is why we as default only show a tree with six levels of branching from the identified outbreak. [Comment only]

See above.

and it is not possible to know how many and which isolates they contain until the tree is viewed.
We have added the information on the number of samples to the website. [Modified code]

This is a welcome addition.

There are only two tree visualisation layouts, it is not possible to subset or colour by metadata, and there is no map, which is a severe limitation. For example, the main competitors in this field (Microreact and Nextstrain), offers all this functionality and more, and are not cited.

Again, we do not see Microreact and Nextstrain as competitors in this field, but rather as very nice visualization tools that would complement PAPABAC very nicely. We have added this to the paper, and we have added a functionality to export data to Microreact. [Modified code, Methods L446-448]

The ability to export to microreact is very welcome here.

In summary, neither the analysis algorithm or visualisation tool are particularly novel, and both have alternatives with more functionality.

This statement is clearly wrong. None of the cited methods have documented ability to classify 160,000+ strains in SNP trees. The only one who comes close is Enterobase which have published that their methods can handle 100,000+ isolates, but that is with cgMLST and not with SNPs. In addition, Evergreen Online can be updated in real-time, which is a feature that, to our knowledge, no other published method has. [Comment only]

Apologies, I was not clear enough here. The ability to process so many isolates and update in real time is technically very impressive. However, I maintain that the algorithm itself is not particularly novel and may cause issues because of the zero distance clustering approach.

However, the ability to search for isolates by SRA ids could be very useful.

It is possible to search for SRA ids. [Comment only]

Apologies, I missed this.

If the underlying trees (both representative of the whole species, and high-resolution trees of individual clades) and metadata could be displayed in Microreact in an automated and continuously updating manner, then this would be a powerful tool which would be very useful. Unfortunately, Evergreen Online in its current state is too limited to be widely useful.

The newick files are already available. As noted above we have now added a functionality to export metadata to Microreact in the form of a csv file. [Modified code, Methods L446-448]

Noted above – this is a great addition.

Specific comments:

P. 2, L. 42: More detail on what else WGS can do (AMR prediction, detect virulence genes etc) is needed here.

This isn't the focus of this manuscript. [Comment only]

Fair enough.

P. 2-3, L. 49-82: Discuss and cite nextstrain, enterobase, pathogen.watch, and microreact

We cited Enterobase, pathogen.watch and Microreact. As we argued before, Nextstrain is primarily for the visualization of viral phylodynamic analysis results, therefore we don't see the relevance of citing it in our manuscript about a bacterial phylogenomic analysis pipeline.

I still think there is enough overlap in approach and audience for it to be worth a brief discussion.

P. 3, Figure 2: the 'ideal' tree on the left has extremely regular branch lengths, compared with ML tree on the right. Why is this? The links between isolates should be shown on this tree, as they are on the supplementary version of the same figure (Figure S2).

This is because it is a tree artificially constructed as it would be if the mutation rate is constant. Links are now displayed in figure. [Updated Figure 2]

Thanks for the clarification.

P. 6, L. 139: A statistical comparison of the trees should be provided.

This has been added to the Results section, L143-145.

Thanks for this.

P. 6, L. 143: This comparison is more relevant than that with the in-vitro experiment, and so I think Figure S3 should be in the main text.

We believe, that the sensitivity of the method is better illustrated with this figure, as the SNP distances between the outbreak, non-outbreak samples are higher in Fig. S3. [Comment only]

Fair enough.

P. 8, L. 184: How were these isolates selected?

This is described in the Methods section, L493-502.

Apologies, I missed this.

P. 8, L. 191: Why were these trees so different? A high resolution ML tree should be inferred for just this clade, as this should help identify which tree (Evergreen or NCBI) is more accurate.

The root cause of the difference between the two trees is the clustering step at 10 SNPs that PAPABAC makes, and the low level genetic diversity in the selected clusters. This is covered in detail in the Discussion, L270-276.

I do not think that this has been adequately addressed. Either the yellow and blue clusters are separate and distinct (PAPABAC), or they are not (NCBI-PD). I understand that the difference in approach has led to different trees, but which is correct? This is important for evaluating both methods. An ML tree should be inferred from only the selected isolates.

P. 10, L. 220-230: Why was 10 SNPs chosen as a threshold? How do the results change if this threshold is changed?

10 SNP was chosen because it is the generally used threshold for being sure that two isolates are from the same outbreak. Increasing the threshold results in the expansion and sometimes merging of the original clusters and the emergence of new clusters. [Comment only]

I think this needs more justification. The appropriate SNP threshold will depend on substitution rate and a number of other factors. See <https://www.biorxiv.org/content/biorxiv/early/2018/12/03/319707> for some discussion of various rates used for TB.

P. 11, L. 271: Again, a high resolution ML tree of the clade should be used to resolve this.

See above.

See response above.

Reviewer #2 (Remarks to the Author):

Review:

Title: Large scale automated phylogenomic analysis of bacterial whole-genome isolates and the Evergreen platform

Authors: Judit Szarvas & Johanne Ahrenfeldt, Jose Luis Bellod Cisneros, Martin Christen Frølund Thomsen, Frank M. Aarestrup, Ole Lund

Summary: The authors present a new phylogenomic pipeline and web platform that performs daily clustering of publicly released pathogen genomes - effectively standing up a genomic surveillance system for a targeted set of pathogen species. This is an enormous task and one that nicely supplements previously released tools within the DTU Center for Genomic Epidemiology. The authors present their pipeline approach and present results from appropriate benchmarking datasets. I would recommend a scientific editor for some text smoothing in areas, but overall the writing is clear and well referenced. This new tool is potentially very valuable to the community - there's only one other comparable tool available that has yet to be published. My review focuses on wanting more clarification around advantages of this new tool, cleaning up some inaccuracies in describing the existing tools, and better targeting the audience so that the reader has a clear understanding of who this tool would benefit most.

Major revisions:

1. After reading this manuscript, I'm left wondering why someone would use the Evergreen platform over NCBI's Pathogen Detection. There seems to be a significant lag for when the raw data is available within Evergreen (e.g. I wasn't able to find any of the data flagged as new within NCBI-PD). I think the authors need to provide more clarity around the advantages of this new pipeline and user interface. Broader set of repositories included? Maybe there is a logical workflow across the larger set of DTU services that makes Evergreen a preferred choice? Is it faster?

2. I think the authors would benefit from more clearly defining the target audience for this tool, then speaking directly to them in this manuscript. Are the target users employees in public health labs? Academic researchers? Epis within Danish public health labs? PulseNet European Epis? COMPARE members? Users of other DTU web services? Once an audience is identified describing a couple use-cases would really help the reader understand its utility. What type of questions are asked and how does Evergreen help answer them? How does evergreen do this better than the NCBI-PD interface?

The NCBI-PD is an unpublished method, and neither is the software publicly available, and thus its accuracy or scientific merit cannot be validated at this time. Being a non NCBI site our speed is limited by the lag before NCBI makes their data available, and the time it takes to reformat the raw reads from NCBI format

to the more commonly used fastq format. That said we have still seen examples such as SRR7280611 which were available in Evergreen Online on June 11, 2018, but not at NCBI-PD before June 13 2018.

In brief, PAPABAC is for anyone who wants/needs open standards, open software, validated methods and possibility of local installation. This is the case for many public health institutions where there may be regulatory requirements for transparency of the methods used. The NCBI-PD do not at the moment fulfill these criteria, but we look forward to see this happening sometime in the future as the reviewer hints to.

It fulfills a need that many employees in public health labs have put to us. Every week new isolates are being sequenced, and they need to know if any of the new isolates are similar to each other or to anything they have ever sequenced before. NCBI-PD may not be a feasible tool for this since many hospitals and public health labs are not allowed to share their data in real time, but they could use the PAPABAC pipeline on their own data to discover local outbreaks.

The main target is in public health labs which sequence more in a continuous mode, rather than scientists that often focus on sequencing a batch of isolates related to a scientific study, but the distinction is blurred for example for scientists doing research in new tools for public health. [Comment only]

3. Table 1 needs some correction/updating:

- *For a wide range of bacterial species. NCBI provides clustering surveillance for 28 pathogen species. What does "wide range" mean here and why doesn't NCBI-PD meet that threshold?*

We were unaware that NCBI have recently expanded to 28 species and have changed the table text to acknowledge this. [Table 1]

- *Requires only raw sequencing reads as input. Correction: NCBI-PD accepts assemblies and raw reads as input.*

We have meant that no additional information or specific configuration was needed in order to run the pipeline. We have amended the text to reflect this. [Table 1]

- *Quality control steps. What are these? Include a description in the methods.*

In short, accepting mapping results higher than 99.0 identity% and a minimum average depth ensures that low quality sequencing runs don't progress further in the pipeline. Coupled that with the significance threshold for the consensus base calling, the consensus sequences are good quality. We have added a description of this to the Method section, L404-412.

- *Communicable nomenclature for subtype and cluster. This appears to be a huge strength of the platform, but as a reader I'm left confused about how Evergreen nomenclature works. Expanding on this feature and contrasting it with other nomenclatures (phylogenetic "addresses" and cgMLST) would really strengthen the manuscript.*

We have expanded the section explaining this. In brief the combination of the tree (template) and the cluster (representative) uniquely identifies a cluster of isolates (which may contain one or more isolates) [Discussion, L237-240]

- *Stable clustering of samples across runs. What does "across runs" mean? Are you making the point that the clusters never change over time? Is this tied to the nomenclature feature above?*

It means, that once a nomenclature template+representative has been assigned to an isolate, it won't

change even if more isolates are added to the tree in subsequent runs. Clusters don't merge, or break up. [Comment only]

4. Would be really nice to have a figure or paragraph outlining dataflow into the Evergreen platform so users understand which isolates they should expect to see in the platform vs. NCBI-PD.

a. Is this correct? Evergreen = all of SRA and ENA. NCBI-PD = SRA data from a list of flagged public-health related bioprojects plus all assembled GenBank genomes.

We have added a description of which isolates have been included, as we think this is better explained in a text than in a figure. [Methods, Online Evergreen platform, L435-439]

b. Describe the QC pipeline implemented within Evergreen and list associated thresholds for inclusion in the clustering analysis. QC is listed in table 1, but not described in methods.

We have expanded the description of our QC in Methods, L404-412

c. Authors say that daily updates are done within Evergreen, but none of the new isolates highlighted within NCBI-PD are present in Evergreen. For example, on March 26 the following new Salmonella SRRs are present in NCBI-PD, but not Evergreen: SRR8775497, SRR8775496, SRR8775495. The salmonella pipeline is the lengthiest for NCBI-PD, so I would expect these to show up in the Evergreen platform sooner if their pipeline is indeed faster.

Our pipeline has always been able to complete the calculations within 24 hours but sometimes the time to download the new isolates have delayed the update of the trees. [Comment only]

5. Because the backbone of this pipeline is NCBI RefSeq's database, I'm curious what the authors think about the representative nature this database. If someone were to design a perfect reference database, you'd want even coverage spanning the diversity of each species. Is this what you see in the references?

We do not show templates that are more than 99% similar, so this evens out the case where there are many very similar templates in the database. [Comment only]

Are there holes in the diversity?

Indeed, there are sequence spaces that are not spanned by complete genomes, and sequencing efforts in these areas should be encouraged. Currently, if a close template is not found for an isolate it is not included. In our experience, clinically important reference genomes generally provide sufficient diversity to cover the sequence space of interest. [Comment only]

If so, what are the effects on the pipeline? Is there any evidence of acquisition bias caused by reference genomes being too distant?

New templates can be added to the database to accommodate this. In principle, each isolate that was not similar to any template can be added to the list of templates, and this will not change any existing trees or nomenclature assignments. [Comment only]

Minor revisions:

Lines 50-57. The summary of GenomeTrakr and NCBI's Pathogen detection isn't quite right here. Consider using the following language.

Since 2012, the US Food and Drug Administration (FDA) has lead a network of public health and university laboratories, called GenomeTrakr. These laboratories sequence bacterial isolates from food and environmental samples and upload raw data to databases at the National Center for Biotechnology Information (NCBI)¹². NCBI's Pathogen Detection portal computes clustering results for 28 pathogenic species, broadly expanding GenomeTrakr's initial foodborne surveillance effort. The NCBI Pathogen Detection pipeline¹³ assembles the samples into draft genomes, scans the current clusters for a within-50-SNP match, then constructs phylogenetic trees for each cluster using an exact maximum compatibility

algorithm14. This approach requires publicly available data from raw or assembled genome submissions and requires extensive computational resources for the larger databases, like Salmonella. No sub-species taxonomical classification has been implemented within this pipeline at this time.

We have edited the paragraph to reflect these facts. [Introduction, L44-53]

Lines 58-71. This paragraph needs some clarification. Setting the MLST scheme discussion aside, the overall dataflow presented here isn't accurate. This sentence is factually incorrect: "Meanwhile, all of the raw data could be kept locally." This sentence might be true for the European members of PulseNet, but it's not accurate for PulseNet USA: "Only data from individual strains would have to be shared when further confirmation of an outbreak is required". All raw PulseNet USA data are uploaded to the NCBI's public Pathogen Detection Pipeline in real-time, where they are clustered with other public surveillance efforts (GenomeTrakr, PHE, etc.), resulting in browsable SNP trees. PulseNet USA also analyzes their data with 3rd party tools using a suite of cgMLST schemes. The data flow within this 3rd party system is separate from the public facing discussion. I think this distinction is really important – the US is united in publicly releasing all raw data

for our foodborne pathogen surveillance. This paragraph makes it seem like there are two separate efforts within the US, which is not the case. Perhaps clarifying US vs. European PulseNet would help here? Or, you can add PN USA to the previous paragraph for the open data effort at NCBI's pathogen detection, then contrast that to the closed system within Europe in the following paragraph.

We meant that it was technically possible to keep data locally, but there may of course be legal requirements to share data. We have revised the text to clarify this. [Introduction, L57-59]

Lines 72-73. I had to re-read this paragraph a couple times to understand the point. I think the authors are saying that analysts at public health labs often view the SNP trees at NCBI and dendograms within BioNumerics as preliminary results. That these results sometimes need further refinement using in-house tools? Perhaps narrowing or expanding taxon sampling, etc. Maybe more clarity in the first sentence would set up the rest of the paragraph? For the most part this further refinement has now become rare in the US as our existing "first pass" tools have become quite good. If this is what the authors meant, how does Evergreen offer this further refinement?

We meant that usually, phylogenetic analysis is done on selected samples, that were assembled, subtyped, phenotyped beforehand. With PAPABAC, phylogenetic analysis can be the first step, as no pre-knowledge about the input data is necessary. Afterwards, the further refinement with in-house methods can be carried out on the clusters that PAPABAC found. We believe, that by performing a bias-free clustering first, more clusters could be detected. [Introduction, L71-73]

Lines 83-90. Making a clearer contrast to NCBI's Pathogen Detection portal and BioNumerics cgMLST would be good to highlight Evergreen's advantages in addition to a stable naming scheme. For example, what are some of the known limitations of current gene- and SNP-based approaches that Evergreen overcomes?

As the introduction section states, selection and curation of cg- and wg-MLST schemes is a limitation to gene-based approaches. Moreover, evolution is ongoing, and calling alleles is not always straightforward process. Reference based SNP-approaches overcome these limitations, but choosing the right reference is crucial. Moreover, with the increasing number of isolates in surveillance, clusters could be overlooked by not including all the isolates involved in an outbreak in a phylogenetic analysis. Our pipeline solves this problem by automatization of the reference selection and isolate-classification based on rapid subtyping. [Comment only]

Lines 152-154: "Raw WGS data files of five major foodborne pathogens (C. jejuni, E. coli, L. monocytogenes, Salmonella enterica, and Shigella spp.) are downloaded daily from public repositories . . .". Please list the public repositories that feed Evergreen its data. For example, is it scanning all of SRA and ENA? Or just a targeted set of BioProjects and/or submitters? How up-to-date is evergreen in comparison to

NCBI-PD? If a set of SRRs found in NCBI-PD are not available in Evergreen is this because there's a lag in updating the system? Or they didn't pass Evergreen QC? Or evergreen monitors a narrower set of repositories than does NCBI-PD?

More details on the inclusion criteria for Evergreen Online have been added, as noted before. [Methods, Online Evergreen platform, L435-439]

Lines 487-490. Figure S3. Please include a description of what the asterisks mean. It appears the PAPABAC pipeline is recovering fewer SNPs, resulting in lower resolution trees within these benchmark datasets (e.g. for the Lm tree NCBI-PD finds about 13 SNP differences between the two major blue OB lineages, but PAPABAC appears to find next to zero based on branch lengths). Is this as expected? Please comment on this observation in the discussion.

Have added the explanation for the asterisk to each figure label. The explanation for the shorter branch lengths, in brief, is that the trees on the left were inferred using the CFSAN SNP pipeline (<https://doi.org/10.7717/peerj-cs.20>), which does extensive filtering on the found SNPs, and collects them into a SNP matrix, that is then the input for the ML method. Therefore it has a higher value for branch length, than PAPABAC does. The zero length branch length inside a cluster is a deliberate choice, as no distances are calculated between the isolated in the cluster. And as the cluster representative was within 10 SNPs of the isolates of the two major lineages, they were merged into one cluster.

Editorial notes:

Title: "phylogenomical". Does this have a different meaning than commonly used word "phylogenomic"? I've never seen this word before and it doesn't turn up any google hits. I would recommend sticking to commonly used terminology unless you require something new, in which case I would recommend defining this new word upfront in the Introduction.

We have changed it to phylogenomic. [Title]

Line 50: GenomeTrackr should be spelled "GenomeTrakr" throughout the manuscript.

We have corrected this. [Introduction]

Line 50-51. "bacterial isolates from clinical and environmental samples..." Should read "...bacterial isolates from food and environmental samples..."

This has been corrected. [Introduction, L45]

Line 303: update link for GenomeTrakr:

<https://www.fda.gov/Food/FoodScienceResearch/WholeGenomeSequencingProgramWGS/default.htm>

Have updated the link. [L307]

Line 309. Curious what this reference is? This recent paper I saw might be a better reference:

Timme RE, Sanchez Leon M, Allard MW 2019. Utilizing the Public GenomeTrakr Database for Foodborne Pathogen Traceback. Methods in molecular biology (Clifton, N.J.) 1918:201–212. DOI: 10.1007/978-1-4939-9000-9_17.

We have changed to cite this reference [L313]

Reviewer #3 (Remarks to the Author):

Dear authors,

This is a well written manuscript that proposes a solution to the rapidly building volume of WGS surveillance data worldwide that are accumulating in silo. The authors have created a bioinformatics pipeline that allows rapid and continuous phylogenomic analysis of large scale genomics data.

This tool is of great value and addresses the gaps in existing databases and tools such as NCBI-PD and Snapper DB. These include allowing for raw sequencing reads as input, being assembly-free, has built in automated phylogenomic analysis, and are open source.

The authors have also addressed the issue of tool validation of the PAPABAC pipeline, and the authors clearly appreciate the importance of cluster generation both locally and across different databases. However, it is not clear whether the raw data can be downloadable for further analysis into the user's server, or whether analysis and comparisons can only be performed on Evergreen. This would be important to clarify.

While I have tried hard to find flaws in this manuscript, I am unable to, and am thrilled to see this tool readily available for WGS surveillance work internationally. I look forward to this tool being a mainstream means for communicating public health related clusters across international boundaries.

I support the publication of this manuscript with minor edits as outlined above.

Reviewers' comments:

Reviewer #1: Sending it as an email attachment

Reviewer #2:

Outstanding revisions:

Lines 58-71. This paragraph needs some clarification. Setting the MLST scheme discussion aside, the overall dataflow presented here isn't accurate. This sentence is factually incorrect: "Meanwhile, all of the raw data could be kept locally." This sentence might be true for the European members of PulseNet, but it's not accurate for PulseNet USA: "Only data from individual strains would have to be shared when further confirmation of an outbreak is required". All raw PulseNet USA data are uploaded to the NCBI's public Pathogen Detection Pipeline in real-time, where they are clustered with other public surveillance efforts (GenomeTrakr, PHE, etc.), resulting in browsable SNP trees. PulseNet USA also analyzes their data with 3rd party tools using a suite of cgMLST schemes. The data flow within this 3rd party system is separate from the public facing discussion. I think this distinction is really important – the US is united in publicly releasing all raw data for our foodborne pathogen surveillance.

This paragraph makes it seem like there are two separate efforts within the US, which is not the case. Perhaps clarifying US vs. European PulseNet would help here? Or, you can add PN USA to the previous paragraph for the open data effort at NCBI's pathogen detection, then contrast that to the closed system within Europe in the following paragraph. We meant that it was technically possible to keep data locally, but there may of course be legal requirements to share data. We have revised the text to clarify this. [Introduction, L57-59]

This still reads as if PulseNet USA and GenomeTrakr have two separate systems (or "visions"), which is not accurate. Foodborne pathogen surveillance in the US might have different analysis approaches (SNP vs. cgMLST) depending on questions being addressed, but the data are all made public together through the NCBI Pathogen Detection portal where they are clustered phylogenetically without any pre-knowledge input. I recommend removing these two statements: "Meanwhile, all of the raw data could be kept locally. Only data from individual strains would have to be shared when further confirmation of an outbreak is required." Or, replace "PulseNet USA" with "PulseNet Europe".

Lines 72-73. I had to re-read this paragraph a couple times to understand the point. I think the authors are saying that analysts at public health labs often view the SNP trees at NCBI and dendrograms within BioNumerics as preliminary results. That these results sometimes need further refinement using in-house tools? Perhaps narrowing or expanding taxon sampling, etc. Maybe more clarity in the first sentence would set up the rest of the paragraph? For the most part this further refinement has now become rare in the US as our existing "first pass" tools have become quite good. If this is what the authors meant, how does Evergreen offer this further refinement?

We meant that usually, phylogenetic analysis is done on selected samples, that were assembled, subtyped, phenotyped beforehand. With PAPABAC, phylogenetic analysis can be the first step, as no pre-knowledge about the input data is necessary. Afterwards, the further refinement with in-house methods can be carried out on the clusters that PAPABAC found. We believe, that by performing a bias-free clustering first, more clusters could be detected. [Introduction, L71-73]

The approaches mentioned above yield preliminary results for outbreak detection, as they often lack the necessary resolution, thus, in most cases, selected WGS data are further analyzed using

single nucleotide profiling” Need to clarify which approaches you are talking about – e.g. this wouldn’t apply to PulseNet USA, mentioned as a previous approach, where the data ARE aggregated in a central public space with phylogenetic clustering as the first step. You could highlight here the utility of PAPABAC for users that aren’t or can’t participate in the NCBI-PD open data model. Or remove the mention of PulseNet USA in the previous paragraph.

Dear Reviewers,

Thank you for your further comments. Please find our responses below.

In short, we answer Reviewer #1's question about the front-end visualization, and point them towards high-resolution trees. We demonstrate that a maximum likelihood tree inferred on the samples in the NCBI comparison set has the same topology as Evergreen Online's high resolution tree, and we have added this refined tree to the manuscript. As for the clustering threshold parameter, we still consider 10 SNPs to be a good arbitrary value, but understand that users might want to customize this setting, therefore it was added to the config file as an option. Moreover, we argue that novelty doesn't require increased complexity.

We also edited Introduction in line with Reviewer #2 requests for clarifications.

In addition, minor stylistic changes were made to the text.

Best regards,
Szarvas et al

Our comments:

Reviewers' comments:

Reviewer #1 (Remarks to the Author):

Szarvas et al. present a pipeline, 'PAPABAC' for the analysis of bacterial genomes, which is paired with a visualisation tool, 'Evergreen Online' for displaying the resulting trees. The aim of this approach is to identify outbreaks, in particular of foodborne pathogens. The pipeline is reference-based, and computational efficiency is improved by removing isolates which are less than 10 SNPs from another isolate, inferring the tree using the 'representative' isolates only, and then placing the removed isolates back onto the tree with 0 distance from the representative isolate for that cluster. New publicly available data from the SRA/ENA is downloaded every day, and the trees are re-inferred to incorporate the new data. The code is open-source, the analysis is sound, and the paper is clear and well written.

The main motivation of the paper, a resource of continuously updating trees of bacterial pathogens, is important, timely, and would be of interest to public health epidemiologists in all settings. However, there are several related tools which are completely omitted from the introduction. Nextstrain (<https://doi.org/10.1093/bioinformatics/bty407>) also displays continuously updated trees of certain pathogens. Although it is mainly focussed on viral pathogens, Tuberculosis is listed on the website. Nextstrain offers more functionality than Evergreen Online, such as more tree visualisation options, an interactive map, and a timeline.

Nextstrain is, as the reviewer notes, focused on viruses, and the 999 Tuberculosis strains covered only goes until August 2018, and is thus not continuously updated. Moreover, its bioinformatics pipeline Augur takes in consensus sequences, not raw sequencing data, and their workflow differs from PAPABAC. We agree that the Nextstrain interface has more interactive features. At the moment NextStrain covers only 9 species, and the largest tree contains 2,255 samples. PAPABAC is designed to handle much larger datasets (160,000 isolates so far). It is unknown if Nextstrain will be able to handle such a large dataset, and if it can, whether the visualizations will be usable for the user. [Comment only]

Regardless of the merits of both approaches, the overlap in scope of the tools means that nextstrain should at least be mentioned.

Fair enough. Added it to Introduction, L90-91.

Microreact (<https://dx.doi.org/10.1099/mgen.0.000093>) offers similar functionality to Nextstrain, but is very easy to upload data to, and has extensive functionality for colouring and labelling isolates by their metadata fields.

We agree again that Microreact has a nice user interface, but the completely comment misses the point that Evergreen Online is developed to handle the analysis and visualization of datasets that are two orders of magnitude larger. And Microreact is not analyzing raw NGS data, but simply visualizing outputs. We have added support to visualize the Evergreen trees via Microreact (<https://microreact.org/upload>) by making the required .tsv and .nwk files available at a publicly available URL under <https://cge.cbs.dtu.dk/services/Evergreen> and via their API. [Modified code, Methods L446-448]

The addition of support for visualisation in microreact is very welcome. Clearly Evergreen is capable of analysing extremely large datasets, however the visualisation tool still struggles with the large trees. For example, the largest tree, “Salmonella_enterica_subsp_enterica_serovar_Typhimurium_str_LT2_NC_003197_2” opens in evergreen, but any attempt to zoom in or click on nodes crashes the browser (on my laptop at least). In contrast, it is possible to interactively view the same tree in microreact (albeit with some lag). Evergreen uses PhyloCanvas version 2.8, while Microreact has the newer, as of yet not released version, that’s optimized for large trees. As soon as the developers release the new version, we’ll perform the upgrade. In the meantime, one can open the subtree that contains a given cluster either via the changes table or the search function. [Comment only]

Enterobase (<https://doi.org/10.1371/journal.pgen.1007261>) continuously downloads publicly available data of foodborne pathogens, but offers far more functionality than Evergreen Online (assemblies, ST typing, sortable metadata).

Enterobase has been cited and the pros and cons of using cgMLST rather than SNPs for phylogeny is extensively discussed. In brief, we believe one of the limitations with cgMLST is the need for a globally agreed and updated allele database for all species to be covered. Furthermore, focusing on the core genome may miss some mutations that are important to track recent evolution. Enterobase is only available for a few bacterial species, whereas Evergreen can cover all bacteria (and potential other) species. [Introduction L54-70]

Agreed about the need for a globally agreed and updated allele database, and focusing on the core means some mutations are missed. However, because the PAPABAC approach is reference based, only SNPs in regions present in the reference will be called, and so some recent mutations may still be missed. The extent to which this happens will also vary based on the distance to the reference.

True, the distance to the reference is minimized by matching each isolate to a template that’s at minimum 99% identical to it. We have also added a sentence stating that reference based method may miss SNPs if they are in regions not covered by the template. [Introduction L80-81]

Pathogen.watch (<https://pathogen.watch>) does not continuously download data, but it plots the samples on an interactive map, does ST typing, predicts antibiotic resistance, in addition to displaying an interactive phylogenetic tree. In contrast, the functionality offered by Evergreen Online is extremely limited - namely a phylogenetic tree to aid detection of outbreaks.

As the reviewer notes pathogen.watch does NOT continuously update with new genomes and this is a very important difference with Evergreen Online. The main accomplishment of the PAPABAC method is that it can update SNP trees with 160.000+ isolates in real time and that we have a hard time seeing as an “extremely limited functionality”. Many of the other phenotype-predictions are already available from our web site cge.cbs.dtu.dk. In the future we may combine these with Evergreen Online, but it is beyond the scope of the current paper. [Introduction L68-69]

Fair enough, the ability to update such large datasets in real time is impressive.

Because this is the only functionality offered by Evergreen Online, it should perform this extremely well. However, it is not clear that this is the case. The algorithm for inferring the trees is reasonable for a quick overview of the population, but does not give enough detail. For example it should be possible to zoom in on a cluster, and re-infer a high-resolution subtree of only these isolates - this would not be computationally demanding, and is vital to provide information on outbreak progression.

We have now added the calculation of a high resolution tree for each cluster with non-zero diversity in Evergreen Online. [Modified code, Methods L441-443]

Where are these trees with non-zero distances? I can only see the original trees with zero distance clusters indicated by asterisks.

They can be reached by the “Refined subtree” link in the phyloCanvas-subtree header. For example:

https://cge.cbs.dtu.dk/services/Evergreen/phyloCanvas.php?templ=Salmonella_enterica_subsp_enterica_serovar_Dublin_str_CT_02021853_NC_011205_1&sid=SRR5441562&full=false
https://cge.cbs.dtu.dk/services/Evergreen/phyloCanvas.php?templ=Listeria_monocytogenes_Finland_1998_NC_017547_1&sid=SRR7172584&full=false

https://cge.cbs.dtu.dk/services/Evergreen/phylocanvas.php?templ=Salmonella_enterica_subsp_enterica_serovar_Typhi_str_CT18_NC_003198_1&sid=SRR5500469&full=false
https://cge.cbs.dtu.dk/services/Evergreen/phylocanvas.php?templ=Escherichia_coli_O157_H7_str_Sakai_chromosome_NC_002695_1&sid=SRR6359226&full=false

We have added a sentence in the paper to highlight this feature. [Results L192-194]

It is also not clear why 99% is used for the initial clustering, as this fragments the species into several sets, each based on a reference genome. This makes it difficult to understand how these trees relate to each other. It would make more sense to have a rougher overall tree for each species, and then be able to zoom in and re-infer a subtree for a potential outbreak cluster (using a member of the cluster as a reference). The rough overall tree could be inferred using mash/fastANI, and this would remove the need to infer the tree using only the reference isolates for each cluster.

We initially tried to find a %ID to correspond to serotypes but found no good correlation. The 99% cluster means for example in E.coli, that isolates will be in the same tree if they have less than approx. 1% times 5 million bases = 50.000 SNPs, meaning that if they are not in the same tree they are definitely not in the same outbreak. We are not convinced that merging all trees into one tree would make it easier for the user to overview the data. We think our solution of making a table shortlisting only those trees where clusters are appearing or expanding will make it easier for epidemiologist to focus on the important areas.

[Comment only]

Agreed about a table shortlisting expanding clusters being the easiest way to flag outbreaks.

However, this is not incompatible with a single overview tree per species, and I respectfully disagree that the current approach is more intuitive.

The suggested approach would mean that the distance calculation using for example mash would be performed on significantly larger sets of data with an $O(n^2)$, that would mean a slower data partitioning than we achieve right now, when we use a down-sampled k-mer and similarity-based method (KMA sparse option) for selecting the references for the examined read files. Not to mention, that a species tree with thousands of isolates wouldn't give better overview, as it would be similarly difficult to browse as our current large trees. Moreover, mash approach wouldn't be feasible to pinpoint outbreak clusters using unassembled reads, as the noise introduced by the sequencing errors skews the calculated mash distance. The error of the calculated mash distance would be similar or higher than the distance itself for genomes more similar than 0.1%, thus selecting samples within, let's say a 100 SNPs of each other would come with high uncertainty.

However, we acknowledge, that it'd be practical to have more information about the templates, therefore a new subpage has been added with typing information on the existing templates. [Modified code]

The web-based visualisation tool is quite clunky and limited. When all trees are listed, there are many for each species,

Yes, there are several trees per species, but given that some of these contains thousands of isolates, we think this is preferable to have one big tree. Even with the number of isolates present in some of the trees, Phylocanvas, which we use to visualize trees becomes very slow, which is why we as default only show a tree with six levels of branching from the identified outbreak. [Comment only]

See above.

and it is not possible to know how many and which isolates they contain until the tree is viewed.

We have added the information on the number of samples to the website. [Modified code]

This is a welcome addition.

There are only two tree visualisation layouts, it is not possible to subset or colour by metadata, and there is no map, which is a severe limitation. For example, the main competitors in this field (Microreact and Nextstrain), offers all this functionality and more, and are not cited.

Again, we do not see Microreact and Nextstrain as competitors in this field, but rather as very nice visualization tools that would complement PAPABAC very nicely. We have added this to the paper, and we have added a functionality to export data to Microreact. [Modified code, Methods L446-448]

The ability to export to microreact is very welcome here.

In summary, neither the analysis algorithm or visualisation tool are particularly novel, and both have alternatives with more functionality.

This statement is clearly wrong. None of the cited methods have documented ability to classify 160,000+ strains in SNP trees. The only one who comes close is Enterobase which have published that their methods can handle 100,000+ isolates, but that is with cgMLST and not with SNPs. In addition, Evergreen Online can be updated in real-time, which is a feature that, to our knowledge, no other published method has.

[Comment only]

Apologies, I was not clear enough here. The ability to process so many isolates and update in real time is technically very impressive. However, I maintain that the algorithm itself is not particularly novel and may cause issues because of the zero distance clustering approach.

The fact that the pipeline's workflow is not overly complicated doesn't mean that it's not novel in this form. There is only NCBI-PD that has the same premise as this pipeline, and that employs different methods (assembly, multiple alignment, exact maximum compatibility algorithm).

We have added subtree inference without the zero distance clustering for each of the outbreaks detected, and added links to these trees on the Evergreen Online web page. In the standalone version the clustering threshold can be modified. [Comment only]

However, the ability to search for isolates by SRA ids could be very useful.

It is possible to search for SRA ids. [Comment only]

Apologies, I missed this.

If the underlying trees (both representative of the whole species, and high-resolution trees of individual clades) and metadata could be displayed in Microreact in an automated and continuously updating manner, then this would be a powerful tool which would be very useful. Unfortunately, Evergreen Online in its current state is too limited to be widely useful.

The newick files are already available. As noted above we have now added a functionality to export metadata to Microreact in the form of a csv file. [Modified code, Methods L446-448]

Noted above – this is a great addition.

Specific comments:

P. 2, L. 42: More detail on what else WGS can do (AMR prediction, detect virulence genes etc) is needed here.

This isn't the focus of this manuscript. [Comment only]

Fair enough.

P. 2-3, L. 49-82: Discuss and cite nextstrain, enterobase, pathogen.watch, and microreact

We cited Enterobase, pathogen.watch and Microreact. As we argued before, Nextstrain is primarily for the visualization of viral phylodynamic analysis results, therefore we don't see the relevance of citing it in our manuscript about a bacterial phylogenomic analysis pipeline.

I still think there is enough overlap in approach and audience for it to be worth a brief discussion.

Included in Introduction, L90-91.

P. 3, Figure 2: the 'ideal' tree on the left has extremely regular branch lengths, compared with ML tree on the right. Why is this? The links between isolates should be shown on this tree, as they are on the supplementary version of the same figure (Figure S2).

This is because it is a tree artificially constructed as it would be if the mutation rate is constant. Links are now displayed in figure. [Updated Figure 2]

Thanks for the clarification.

P. 6, L. 139: A statistical comparison of the trees should be provided.

This has been added to the Results section, L143-145.

Thanks for this.

P. 6, L. 143: This comparison is more relevant than that with the in-vitro experiment, and so I think Figure S3 should be in the main text.

We believe, that the sensitivity of the method is better illustrated with this figure, as the SNP distances between the outbreak, non-outbreak samples are higher in Fig. S3. [Comment only]

Fair enough.

P. 8, L. 184: How were these isolates selected?

This is described in the Methods section, L493-502.

Apologies, I missed this.

P. 8, L. 191: Why were these trees so different? A high resolution ML tree should be inferred for just this clade, as this should help identify which tree (Evergreen or NCBI) is more accurate.

The root cause of the difference between the two trees is the clustering step at 10 SNPs that PAPABAC makes, and the low level genetic diversity in the selected clusters. This is covered in detail in the Discussion, L270-276.

I do not think that this has been adequately addressed. Either the yellow and blue clusters are separate and distinct (PAPABAC), or they are not (NCBI-PD). I understand that the difference in approach has led to different trees, but which is correct? This is important for evaluating both methods. An ML tree should be inferred from only the selected isolates.

As these samples are not from simulated data, their “true” phylogeny is not known, so we can’t tell which tree is more correct, but a maximum likelihood tree could indeed be taken as the reference tree.

The ML tree on the top left figure below was inferred on the 440 samples that were selected originally for the comparison with NCBI-PD. The CFSAN pipeline was used for SNP calling, and the resulting SNP matrix with 13184 columns (6623 parsimony-informative sites) was the input to IQ-tree with GTR+G nucleotide substitution model for tree inference. The displayed tree is a subtree with 180 taxa. On the top right is the refined subtree containing all the labeled samples, from

https://cge.cbs.dtu.dk/services/Evergreen/phylocanvas.php?templ=Escherichia_coli_O157_H7_str_Sakai_chromosome_NC_002695.1&sid=SRR6766978&full=false, downloaded on the 30th of October, with additional samples that weren’t pruned from the tree. Based on visual inspection, the tree topologies agree on the selected clades.

The pair of trees below are the same phylogenies, colored using the color scheme in the paper.

On the refined tree around SRR6766978, the blue and yellow labeled samples are also intermixing, like on the NCBI-PD tree. The SNP distance between them are less than 10 (the refined trees have a stricter filtering for unknown bases), so they are all (presumably) part of the same outbreak.

In general, this analysis show that it is only when isolates are closer than the clustering threshold, that NCBI-PD and EO disagree, which means that for most purposes they are equally useful for outbreak analysis.

We added the refined subtree, that was pruned to remove the “newer” taxa, to the supplementary as Fig. 5, and referred it in Results L230-231 and Discussion L 311-312. The common sample IDs from NCBI-PD and EO were added in Supplementary_data1.xls.

2.0E-4

3.0

P. 10, L. 220-230: Why was 10 SNPs chosen as a threshold? How do the results change if this threshold is changed?

10 SNP was chosen because it is the generally used threshold for being sure that two isolates are from the same outbreak. Increasing the threshold results in the expansion and sometimes merging of the original clusters and the emergence of new clusters. [Comment only]

I think this needs more justification. The appropriate SNP threshold will depend on substitution rate and a number of other factors. See <https://www.biorxiv.org/content/biorxiv/early/2018/12/03/319707> for some discussion of various rates used for TB.

We agree, that for each species (even subtype), the applicable SNP threshold will depend on number of factors. Unfortunately, majority of public data lacks exact sampling date, therefore a probabilistic solution such as in the cited article can't be used. For hard thresholds, <https://www.ncbi.nlm.nih.gov/pmc/articles/PMC6048267/> and [https://www.clinicalmicrobiologyandinfection.com/article/S1198-743X\(17\)30710-3/pdf](https://www.clinicalmicrobiologyandinfection.com/article/S1198-743X(17)30710-3/pdf) list some examples of relatedness criteria based on studied outbreaks. 10 SNPs is on the lower range for these, and could be considered a good approximation in lieu of a custom threshold. Furthermore, for foodborne pathogens, it is recommended, to consider not just SNP distances and tree topology, but also epidemiological data and traceback data, when establishing linkage between isolates (<https://www.ncbi.nlm.nih.gov/pmc/articles/PMC6048267/>). This is also stressed in the Discussion L267-269.

This parameter can be modified by the user, now more easily in the config.py. [Modified code]

P. 11, L. 271: Again, a high resolution ML tree of the clade should be used to resolve this.

See above.

See response above.

See response above.

Reviewer #2 (Remarks to the Author):

Outstanding revisions:

Lines 58-71. This paragraph needs some clarification. Setting the MLST scheme discussion aside, the overall dataflow presented here isn't accurate. This sentence is factually incorrect: "Meanwhile, all of the raw data could be kept locally." This sentence might be true for the European members of PulseNet, but it's not accurate for PulseNet USA: "Only data from individual strains would have to be shared when further confirmation of an outbreak is required". All raw PulseNet USA data are uploaded to the NCBI's public Pathogen Detection Pipeline in real-time, where they are clustered with other public surveillance efforts (GenomeTrakr, PHE, etc.), resulting in browsable SNP trees. PulseNet USA also analyzes their data with 3rd party tools using a suite of cgMLST schemes. The data flow within this 3rd party system is separate from the public facing discussion. I think this distinction is really important – the US is united in publicly releasing all raw data for our foodborne pathogen surveillance.

This paragraph makes it seem like there are two separate efforts within the US, which is not the case. Perhaps clarifying US vs. European PulseNet would help here? Or, you can add PN USA to the previous paragraph for the open data effort at NCBI's pathogen detection, then contrast that to the closed system within Europe in the following paragraph.

We meant that it was technically possible to keep data locally, but there may of course be legal requirements to share data. We have revised the text to clarify this. [Introduction, L57-59]

****This still reads as if PulseNet USA and GenomeTrakr have two separate systems (or "visions"), which is not accurate. Foodborne pathogen surveillance in the US might have different analysis approaches (SNP vs. cgMLST) depending on questions being addressed, but the data are all made public together through the NCBI Pathogen Detection portal where they are clustered phylogenetically without any pre-knowledge input. I recommend removing these two statements: "Meanwhile, all of the raw data could be kept locally. Only data from individual strains would have to be shared when further confirmation of an outbreak is required." Or, replace "PulseNet USA" with "PulseNet Europe".**

We removed the sentences: "Meanwhile, all of the raw data could be kept locally. Only data from individual strains would have to be shared when further confirmation of an outbreak is required." [Introduction L57]

Lines 72-73. I had to re-read this paragraph a couple times to understand the point. I think the authors are saying that analysts at public health labs often view the SNP trees at NCBI and dendograms within BioNumerics as preliminary results. That these results sometimes need further refinement using in-house tools? Perhaps narrowing or expanding taxon sampling, etc. Maybe more clarity in the first sentence would set up the rest of the paragraph? For the most part this further refinement has now become rare in the US as our existing "first pass" tools have become quite good. If this is what the authors meant, how does Evergreen offer this further refinement?

We meant that usually, phylogenetic analysis is done on selected samples, that were assembled, subtyped, phenotyped beforehand. With PAPABAC, phylogenetic analysis can be the first step, as no pre-knowledge about the input data is necessary. Afterwards, the further refinement with in-house methods can be carried out on the clusters that PAPABAC found. We believe, that by performing a bias-free clustering first, more clusters could be detected. [Introduction, L71-73]

*****"The approaches mentioned above yield preliminary results for outbreak detection, as they often lack the necessary resolution, thus, in most cases, selected WGS data are further analyzed using single nucleotide profiling" Need to clarify which approaches you are talking about – e.g. this wouldn't apply to PulseNet USA, mentioned as a previous approach, where the data ARE aggregated in a central public space with phylogenetic clustering as the first step. You could highlight here the utility of PAPABAC for users that aren't or can't participate in the NCBI-PD open data model. Or remove the mention of PulseNet USA in the previous paragraph.**

We clarified that we meant the gene based approaches. [Introduction L75]

REVIEWERS' COMMENTS:

Reviewer #2 (Remarks to the Author):

I have reviewed the author's responses to both reviewers and feel that all major and minor comments have been thoroughly addressed in the manuscript and in the rebuttal letter.